# Physics-Learning AI Datamodel (PLAID) datasets: a collection of physics simulations for machine learning

## Abstract

Machine learning-based surrogate models have emerged as a powerful tool to accelerate simulation-driven scientific workflows. However, their widespread adoption is hindered by the lack of large-scale, diverse, and standardized datasets tailored to physics-based simulations. While existing initiatives provide valuable contributions, many are limited in scope-focusing on specific physics domains, relying on fragmented tooling, or adhering to overly simplistic datamodels that restrict generalization. To address these limitations, we introduce PLAID (Physics-Learning AI Datamodel), a flexible and extensible framework for representing and sharing datasets of physics simulations. PLAID defines a unified standard for describing simulation data and is accompanied by a library for creating, reading, and manipulating complex datasets across a wide range of physical use cases (gitlab.com/drti/plaid). We release six carefully crafted datasets under the PLAID standard, covering structural mechanics and computational fluid dynamics, and provide baseline benchmarks using representative learning methods. Benchmarking tools are made available on Hugging Face, enabling direct participation by the community and contribution to ongoing evaluation efforts (huggingface.co/PLAIDcompetitions).

## 1 Introduction

Numerical simulation is a cornerstone of scientific and engineering research, providing essential insights into complex physical phenomena across a wide range of domains—including earth and environmental sciences [1], life sciences and medicine [2], finance and economics [3], and industrial engineering [4, 5, 6]. These simulations rely on solving partial differential equations (PDEs) using space and time discretization and numerical methods, typically implemented in large-scale computational solvers. While accurate, these simulations are often computationally intensive, with a single high-fidelity run potentially requiring several hours or days. Many practical scenarios demand solving the same physical model across a wide range of settings—such as in design exploration, optimization, real-time simulation, and uncertainty quantification. In such many-query contexts, reliance on costly simulations becomes impractical. To address this, a broad spectrum of surrogate modeling techniques has been proposed to approximate simulation outputs at a fraction of the computational cost.

Classical surrogate models perform non-linear regression over parametric spaces using statistical learning techniques, such as polynomial regression, nearest neighbors, support vector machines, random forests [7], and Gaussian processes [8]. These models are widely supported by software libraries such as UQLab [9], OpenTURNS [10], Dakota [11] and Lagun [12]. However, they are typically restricted to low-dimensional, tabular parameter spaces and cannot be directly used in more complex simulation setups. In contrast, many modern applications involve richer input configurations, including unstructured geometries, spatially varying fields, and complex boundary or

material conditions. These settings require learning from heterogeneous, high-dimensional data with nonparametric variability.

Recent advances in scientific machine learning have begun to address these challenges. One line of work, often referred to as physics-based model reduction, builds surrogates that approximate the solution of the governing equations directly [13, 14, 15, 16, 17, 18]. Other approaches have also been proposed using non-parametric methods based on the use of morphing [19, 20, 21] or optimal transport [22, 23], and have the advantage of requiring a smaller number of design points. Increasingly, deep learning methods—particularly Graph Neural Networks (GNNs)—have shown promise in capturing the spatiotemporal dynamics of physical systems. Building on the message-passing paradigm introduced in [24], architectures such as MeshGraphNets [25] extend GNNs to general mesh-based simulations. Hierarchical versions like MultiScale MeshGraphNets [26] enhance scalability and accuracy, while recent works demonstrate effectiveness in inverse [27] and steady-state problems [28]. Other developments include geodesic convolutions [29], multi-resolution models [30, 31], and improved pooling strategies [32]. Tools such as PhysicsNeMo [33], PyTorch Geometric [34], and Deep Graph Library [35] provide convenient foundations for implementing these methods.

Despite these advances, widespread adoption remains hindered by a critical bottleneck: the lack of large-scale, diverse, and standardized datasets for training and benchmarking. Existing datasets often cover narrow physical regimes, rely on ad hoc formats, or are tied to specific libraries—limiting reusability and interoperability. Furthermore, many datasets are tailored to isolated challenges (e.g., time dependence) but fail to accommodate others (e.g., geometric variation). This fragmentation is particularly problematic in the context of recent developments in physics foundation models [36, 37, 38, 39, 40], which require large, flexible, and standardized sources of training data.

To address these limitations, we introduce PLAID (Physics-Learning AI Datamodel), a comprehensive framework for representing and manipulating datasets of physics simulations for machine learning. PLAID defines a generic, extensible datamodel that supports a wide range of use cases—including time-dependent problems, remeshing, mixed-element unstructured meshes, node/element tagging, multiple spatial dimensions and topologies. We provide an accompanying software library to facilitate dataset creation, reading, and high-level interaction, that can leverage Hugging Face infrastructure for efficient streaming, caching, and sharing.

In Section 2, we review relevant dataset efforts in the literature. Section 3 introduces the PLAID datamodel and implementation, along with six publicly released datasets in structural mechanics and computational fluid dynamics, presented in Section 4, that showcase rich variability in physics and numerical complexity. In Section 5, we provide performance benchmarks across a range of machine learning methods, hosted on Hugging Face to allow community participation and continual updates. We conclude with perspectives in Section 6.

## 2 Related Work

Progress in machine learning has been largely driven by the availability of large, diverse, and carefully curated datasets [41, 42, 43]. Natural language processing models are trained on web-scale data [44, 45, 46, 47], and vision models routinely leverage billions of image–text pairs [48, 49, 50].

In contrast, datasets for physics learning remain comparatively underdeveloped. Early benchmarks targeted standard physics problems and reference simulations [51, 52, 53]. More recent datasets have focused on complex, domain-specific settings [54, 55, 56, 57, 58, 59, 60, 61, 62]. The recently proposed Well [63] includes an impressive list of datasets for various physics, but is limited to structured grids (uniformly sampled domains).

Structural mechanics simulations, with non-linear constitutive laws, are of paramount importance for industrial design, and are poorly represented in available datasets. Most available datasets use a datamodel that limit their evolution and generality. Complex industrial settings include vertices and element tags, heterogeneous data with remeshing, multiple meshes of various dimensions, topologies and mixed element types, compatible with commercial codes routinely used by design engineers. Besides, most datasets come with a library dedicated to the dataset, featuring specific commands and hypothesis, which limit they wide adoption.

## 3  PLAID standard

We propose PLAID, a datamodel for datasets for machine learning applied to physics-related problems.
PLAID also refers to the library that implements this datamodel, available on GitLab [64], and to
the file format used to store data. The primary goal of the library is to provide a general and
flexible framework for defining physics-based dataset, along with a corresponding learning task. The
datamodel is built on CGNS [65], leveraging its well-established formalization of a wide range of
physical configurations.

PLAID datasets are provided either as human-readable data storage, or stored using Hugging Face
datasets tools [66]. In the former case, YAML and CSV files can be opened with any text editor,
while CGNS files containing physical configurations can be visualized using tools such as ParaView,
see Figure 1. In the latter, we benefit directly from powerful data management such as caching and
online streaming.

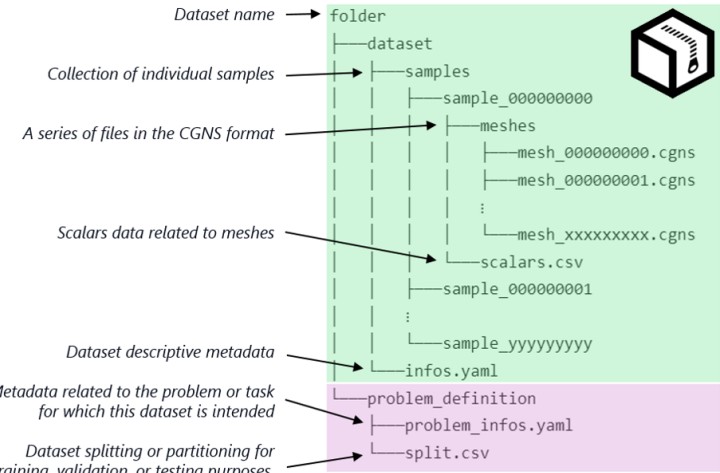

**Figure 1:** PLAID files structure.

Additionally, PLAID offers high-level utilities for constructing, handling and read/write datasets
efficiently. Documentation is available online, with usage examples and tutorials showing how one
can create a PLAID dataset from its own data. We also mention Muscat, a finite element toolbox
available on GitLab [67, 68], containing various reader and writers from and to various files formats
used in numerical simulation codes for physics, and routines to generate the CGNS data structures
used in PLAID. Samples can feature multiple meshes, scalars, fields and time series. We illustrate
how PLAID can deal with complex heterogeneous data by explaining some available commands:

- `dataset[0].get_field_names(name, zone_name, base_name, location, time)`: retrieves the first sample's field called `name`, for chosen `zone_name`, `base_name`, `location` (Vertex, CellCenter, FaceCenter, ...) and `time` in the CGNS structure. Fields and meshes can change over time, allowing remeshing and field appearance/disappearance at any time step.

- `sample.get_field(name)`: automatic handling of default values to prevent exposing `zone_name`, `base_name` and `location` arguments to simple cases with no ambiguity.

- `sample.get_mesh(apply_links = True)`: allows to link meshes between CGNS data structures to prevent multiple copies in case of fixed mesh cases.

More examples are provided in Appendix B.

## 4 PLAID datasets

### 4.1 Structural mechanics

#### 4.1.1 `Tensile2d` [69] (Zenodo, Hugging Face)

`Tensile2d` is a simple dataset of 2D quasistatic non-linear structural mechanics simulations, in small deformations and plane strain regimes, solved with Z-set [70] using the finite element method. The material is modeled with a non-linear constitutive law. The dataset computes the deformation of a structure subjected to an imposed negative constant pressure at the top, and zero displacement at the bottom, see Figure 2 (left). Only the steady-state solution is kept.

Input variability in the dataset are the unstructured meshes (variable shape, number of nodes and connectivity), the pressure P at the top boundary condition (scalar) and 5 scalars modeling the non-linear constitutive law: (p1, p2, p3, p4 and p5). Outputs of interest are 4 scalars (`max_von_mises`, `max_q`, `max_U2_top` and `max_sig22_top`) and 6 fields (U1, U2, q, sig11, sig22 and sig12). Seven nested training sets are provided, as well as a testing set and two out-of-distribution samples.

#### 4.1.2 `2D_MultiScHypEl` [71] (Zenodo, Hugging Face)

`2D_MultiScHypEl`, standing for 2D multiscale hyperelasticity, is a dataset of 2D quasistatic non-linear structural mechanics simulations under large deformation and plane strain conditions, solved with DOLFINx [72] using the finite element method. The material behavior follows a compressible hyperelastic constitutive law, capturing complex non-linear responses. Each simulation corresponds to the homogenization of a porous representative volume element (RVE), subject to kinematically uniform boundary conditions (KUBC) [73], see Figure 2 (right).

Input variability in the dataset are the unstructured meshes (variable shape, number of nodes, connectivity and topology–the number of circular inclusions) and the 3 scalars modeling the KUBC, namely the components C11, C12, and C22 of the macroscopic right Cauchy-Green deformation tensor. Outputs of interest are 1 scalar (`effective_energy`) and 7 fields (displacements u1, u2; first Piola-Kirchhoff stress components P11, P12, P22, P21 and the strain energy density field `psi`). Various training and testing sets are provided (both across all topologies and within each topology class).

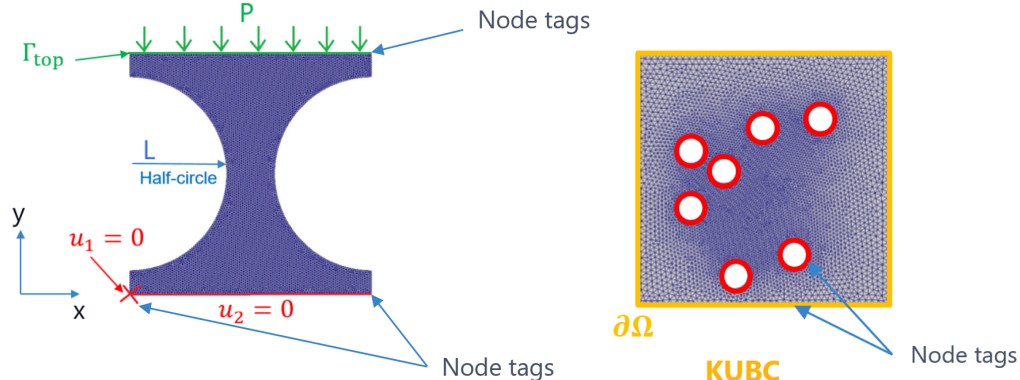

**Figure 2:** Settings for `Tensile2d` (left) and `2D_MultiScHypEl` (right).

#### 4.1.3 `2D_ElPlDynamics` [74] (Zenodo, Hugging Face)

`2D_ElPlDynamics`, standing for 2D elasto-plasto dynamics, is a dataset of 2D dynamic non-linear structural mechanics simulations, in large deformations and plane strain regimes, solved with Open-Radioss [75] using the finite element method. The material is modeled with a non-linear elastoplastic law, with damage (modeled using element erosion), failure and a non-local method for reducing mesh sensitivity. The dataset computes the transient deformation of a 2D structure, subjected to imposed displacement on the right and zero displacement on the left, see Figure 3 (left).

Input variability in the dataset are the unstructured meshes (variable shape, number of nodes, connectivity and topology). Outputs of interest are 3 fields (U_x and U_y the displacement fields at the

nodes, and `EROSION_STATUS` a boolean field at the elements describing the state – valid or broken – of each element). A training and a testing set are provided.

## 4.2 Computational fluid mechanics

### 4.2.1 `Rotor37` [76] (Zenodo, Hugging Face)

`Rotor37` is a dataset of 3D compressible steady-state Reynolds-Averaged Navier-Stokes (RANS) simulations, solved with elsA [77] using the finite volumes method. Large scale simulations around the rotor37 blade inside a 3D duct have been computed, with inflow, outflow and periodic boundary conditions. An adequate turbulence model and laws of the wall have been chosen. The dataset only keeps the steady-state solution at the boundary of the blade inside the duct, and scalars of interest, see Figure 3 (right).

Input variability in the dataset are the block-structured anisotropic meshes (variable shape, normals at the blade surface are provided) and 2 scalars (the pressure `P` and the rotation speed `Omega` of the blade). Outputs of interest are 3 scalars (`Massflow`, `Compression_ratio` and `Efficiency`) and 3 fields (`Density`, `Pressure`, `Temperature`). Eight nested training sets and a testing set are provided.

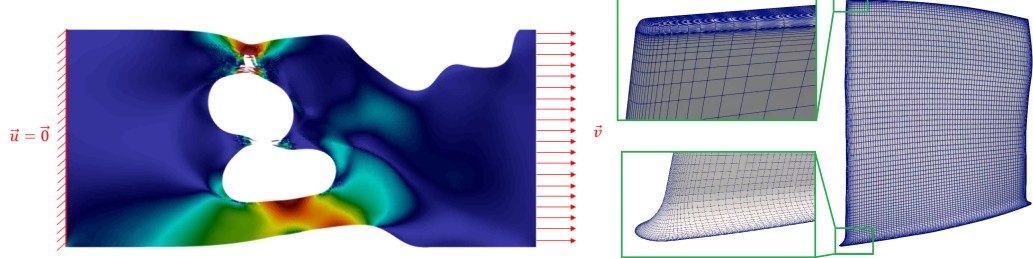

**Figure 3:** Settings for `2D_ElPlDynamics` (left) and `Rotor37` (right).

### 4.2.2 `2D_profile` [78] (Zenodo, Hugging Face)

`2D_profile` is a dataset of 2D compressible steady-state Reynolds-Averaged Navier-Stokes (RANS) simulations, solved with elsA [77] using the finite volumes method. The flow is computed around 2D profiles, which present large deformation around shapes resembling airfoils or propeller blades, on a large refined meshes, with inflow, outflow and periodic boundary conditions, at a supersonic regime. An adequate turbulence model and laws of the wall have been chosen. The dataset only keeps the steady-state solution on a zone cropped close to the profile, see Figure 4 (left).

Input variability in the dataset are the unstructured anisotropic meshes (variable shape, number of nodes and connectivity). Outputs of interest are 4 fields (`Mach`, `Pressure`, `Velocity-x` and `Velocity-y`). A training and a testing set are provided.

### 4.2.3 `VKI-LS59` [79] (Zenodo, Hugging Face)

`VKI-LS59` is a dataset of 2D compressible steady-state Reynolds-Averaged Navier-Stokes (RANS) simulations, solved with BROADCAST [80] using the finite volumes method with high-order corrections. The flow is computed around the VKI-LS59 blade, with inflow, outflow and periodic boundary conditions. A Spalart-Allmaras turbulence model has been chosen, see Figure 4 (right).

Input variability in the dataset are the block-structured anisotropic meshes (variable shape, number of nodes and connectivity, the distance field to the blade surface is provided) and 2 scalars (`angle_in` and `mach_out`). Outputs of interest are 6 scalars (`Q`, `power`, `Pr`, `Tr`, `eth_is` and `angle_out`) and 7 fields (`ro`, `rou`, `rov`, `roe`, `nut`, `mach` and `M_iso` – this last being only defined at the surface of the blade). Eight nested training sets are provided, as well as a testing set.

### 4.2.4 `AirfRANS` [55]

`AirfRANS` is a dataset of external aerodynamics, featuring steady-state Reynolds-Averaged Navier-Stokes (RANS) simulations over airfoils at a subsonic regime, proposed in [55], which we refer to for

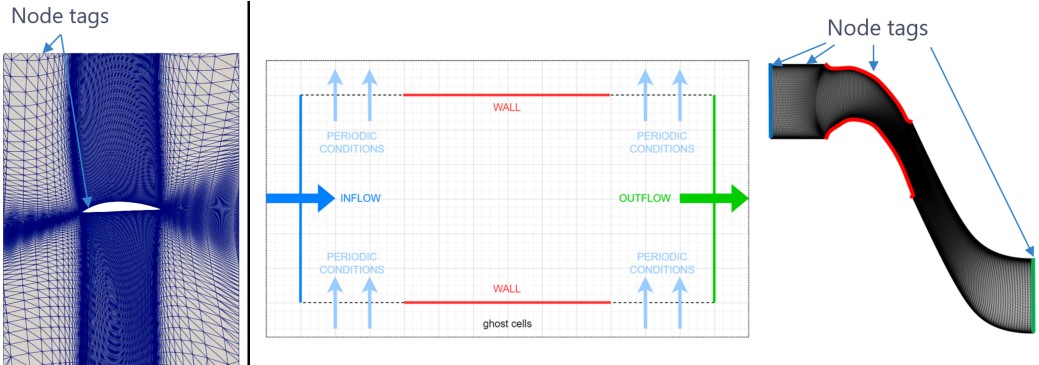

**Figure 4:** Settings for `2D_profile` (left) and `VKI-LS59` (right).

a detailed description. In addition to the six original datasets, we provide three variants of `AirfRANS` in PLAID format: original [81](Zenodo, Hugging Face), clipped [82](Zenodo, Hugging Face) and remeshed [83](Zenodo, Hugging Face).

Input variability in the dataset are the anisotropic meshes (variable shape, number of nodes and connectivity, the distance field to the airfoil surface is provided) and 2 scalars (`angle_of_attack` and `inlet_velocity`). Outputs of interest are 2 scalars (`C_D` and `C_L`) and 4 fields (`nut`, `Ux`, `Uy` and `p`). Various training and testing sets are provided.

## 4.3 Dataset collection

The collection of proposed datasets is available online in a Zenodo community and a Hugging Face community. Description summaries are provided in Tables 1 and 2. The collection will be enriched in the future with additional datasets. Since these datasets have been constructed with the goal to provide open benchmarks to the community, the outputs are not provided on the testing sets – but we provide tools to evaluate scores on these testing sets. Some field outputs are illustrated in Table 3.

| Dataset | Simulation code | Model | Nb samples | Volume Zenodo | Volume HF |
|---|---|---|---|---|---|
| Tensile2d | Z-set | 2D quasistatic non-linear structural mechanics, small deformations, non-linear constitutive law | 702 | 290 MB | 383 MB |
| 2D_MultiScHypEl | DOLFINx | 2D quasistatic non-linear structural mechanics, finite elasticity | 1,140 | 350 MB | 419 MB |
| 2D_ElPlDynamics | OpenRadioss | 2D dynamic non-linear structural mechanics, non-linear non-local constitutive law | 1,018 | 5.7 GB | 8.6 GB |
| Rotor37 | elsA | 3D Navier-Stokes (RANS) | 1,200 | 3.3 GB | 4.0 GB |
| 2D_profile | elsA | 2D Navier-Stokes (RANS) | 400 | 660 MB | 814 MB |
| VKI-LS59 | BROADCAST | 2D Navier-Stokes (RANS) | 839 | 1.9 GB | 2.3 GB |
| AirfRANS original | | | | 9.3 GB | 15.6 GB |
| AirfRANS clipped | OpenFOAM | 2D Navier-Stokes (RANS) | 1,000 | 9.7 GB | 18.2 GB |
| AirfRANS remeshed | | | | 520 MB | 611 MB |

**Table 1:** Dataset collection description: model and simulation volume.

## 5 Benchmark

We first mention that we do not provide benchmark tools and results for the `AirfRANS` datasets, since outputs are public on the testing sets, and various benchmarks are already available in articles [19, 55] and in a competition at NeurIPS 2024 [84].

| Dataset | Mesh (mean nodes) | Inputs | Outputs | Splits (train/test) |
|---|---|---|---|---|
| Tensile2d | tri (9,428) | mesh, 6 scalars | 4 scalars, 6 fields | 500 / 200 |
| 2D_MultiScHypEl | tri (5,692) | mesh, 3 scalars | 1 scalar, 7 fiels | 764 / 376 |
| 2D_ElPlDynamics | tri (25,429) | mesh | 3 fields | 1,000 / 18 |
| Rotor37 | quad (29,773*) | mesh, 2 scalars | 4 scalars, 3 fields | 1,000 / 200 |
| 2D_profile | tri (37,042) | mesh | 4 fields | 300 / 100 |
| VKI-LS59 | quad (36,421*) | mesh, 2 scalars | 6 scalars, 7 fields | 671 / 168 |
| AirfRANS original | quad (179,776) | | | |
| AirfRANS clipped | tri (179,779) | mesh, 2 fields | 2 scalars, 4 fields | various splits |
| AirfRANS remeshed | tri (7,624) | | | |

**Table 2:** Dataset collection description: data and splits, a ∗ in the second column means that the number of nodes and connectivity are constant in the dataset – the position of the nodes still varies.

| Dataset | Examples of field outputs |
|---|---|
| | 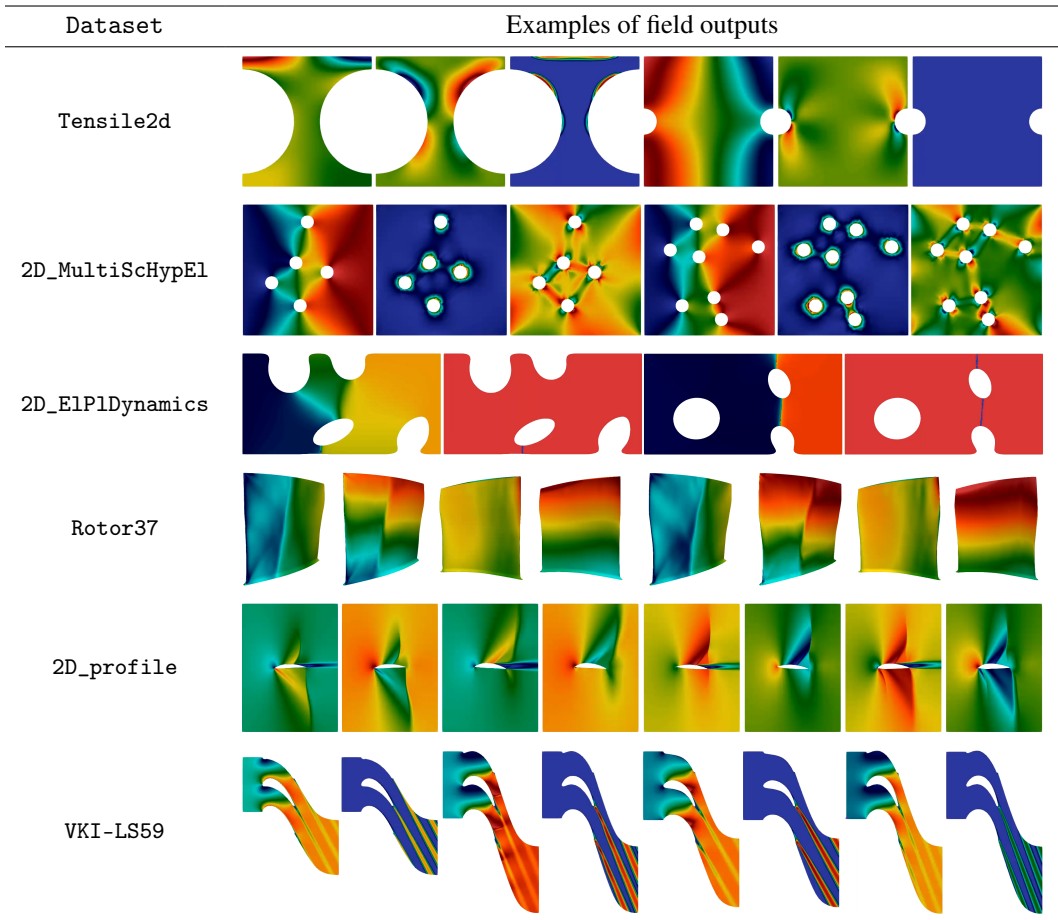 |

**Table 3:** Dataset collection examples of field outputs illustrations.

## 5.1 Methods

For practical reasons, we limit the benchmarks to the few following methods:

- MeshGraphNets (MGNs) [25] are GNNs that utilize an encode-process-decode architecture, transforming mesh data into graph structures, processing them through message passing, and decoding the results to predict field outputs.

- Mesh Morphing Gaussian Processes (MMGP) [19] rely on mesh morphing, finite element interpolation and dimensionality reduction to pretreat mesh-based data into a low dimensional embedding, and utilizes Gaussian processes to predict output scalars and fields.

- Vi-Transformer [85] and Augur [1] rely on mesh partitioning to build tokens related to local mesh information and utilize a transformer to predict scalar and field outputs.

- Domain Agnostic Fourier Neural Operators (DAFNO) [86] handle problems involving irregular geometries and evolving domains. It incorporates a smoothed characteristic function into the integral layer architecture of FNOs, allowing the use of Fast Fourier Transform (FFT) for efficient computations while explicitly encoding geometric information.

- Modulated Aerodynamic Resolution Invariant Operator (MARIO) [87] builds upon [88] and exploits implicit neural representations, which model continuous signals by mapping input coordinates directly to output values, without relying on discrete grids or explicit storage.

For more details on the methods and their respective advantages/drawbacks, refer to Appendix A.

## 5.2 Evaluation metric

Accuracy of the trained models is evaluated by computing RRMSEs (Relative Root Mean Square Errors). Let $\{\mathbf{f}_{\mathrm{ref}}^i\}_{i=1}^{n_\star}$ and $\{\mathbf{f}_{\mathrm{pred}}^i\}_{i=1}^{n_\star}$ be respectively the reference and prediction of a field output on the testing set. The RRMSE is defined as

$$\mathrm{RRMSE}_f(\mathbf{f}_{\mathrm{ref}}, \mathbf{f}_{\mathrm{pred}}) = \left( \frac{1}{n_\star} \sum_{i=1}^{n_\star} \frac{\frac{1}{N^i} \|\mathbf{f}_{\mathrm{ref}}^i - \mathbf{f}_{\mathrm{pred}}^i\|_2^2}{\|\mathbf{f}_{\mathrm{ref}}^i\|_\infty^2} \right)^{1/2},$$

where $N^i$ is the number of nodes in the mesh of sample $i$, $n_\star$ is the number of samples in the testing set, and $\|\mathbf{f}_{\mathrm{ref}}^i\|_\infty$ is the maximum component in the vector $\mathbf{f}_{\mathrm{ref}}^i$. Similarly for scalar outputs, the following relative RMSE is computed:

$$\mathrm{RRMSE}_s(\mathbf{s}_{\mathrm{ref}}, \mathbf{s}_{\mathrm{pred}}) = \left( \frac{1}{n_\star} \sum_{i=1}^{n_\star} \frac{|s_{\mathrm{ref}}^i - s_{\mathrm{pred}}^i|^2}{|s_{\mathrm{ref}}^i|^2} \right)^{1/2}.$$

The score of a submission, `total_error`, is the mean over fields and scalars RRMSEs.

## 5.3 Benchmark results

All individual RRMSE and `total_error` for each method applied to each dataset are reported in Table 4. These results are considered neither exhaustive, nor definitive.

We provide the community with online benchmarking applications hosted on Hugging Face as competitions with no end date, see Hugging Face benchmark collection. Each benchmark comes with a visualization application of the datasets, a description of inputs and outputs and detailed instructions for accessing the data and constructing a prediction file. Anyone can register and submit a prediction: submissions are automatically ranked based on `total_error` as defined in Section 5.2. Hence, the benchmark results presented here will be updated in the future. See Section C for additional details on the benchmarking applications.

We notice that MMGP, Vi-Transformer/Augur and MARIO models perform particularly well on our steady-state datasets, while DAFNO has only been evaluated on our unique time-dependent dataset.

# 6 Conclusion and perspectives

**Limitations.** PLAID is designed to accommodate a wide range of complex scenarios and remains adaptable to emerging use cases that may not be fully addressed by the current datamodel. We plan to expand our collection with more diverse and large-scale datasets of industrial relevance, complemented by benchmarking applications accessible to the community.

**Roadmap.** Future developments include the creation of generic PyTorch dataloaders for PLAID, and the standardization of evaluation metrics and training/inference pipelines based on PLAID.

---

[1]commercial solution from Augur company

| Field, *scalar* output | MGN | MMGP | Vi-Transf. | Augur | DAFNO | MARIO |
|---|---|---|---|---|---|---|
| **Tensile2d** | | | | | | |
| **U1** | 0.0788 | **0.0016** | 0.0344 | 0.0093 | - | - |
| **U2** | 0.1237 | **0.0013** | 0.0424 | 0.0135 | - | - |
| **sig11** | 0.1726 | **0.0037** | 0.0715 | 0.0187 | - | - |
| **sig22** | 0.0560 | **0.0015** | 0.0341 | 0.0099 | - | - |
| **sig12** | 0.0570 | **0.0026** | 0.0494 | 0.0121 | - | - |
| *max_von_mises* | 0.0185 | **0.0050** | 0.0145 | 0.0219 | - | - |
| *max_U2_top* | 0.0292 | **0.0042** | 0.0210 | 0.0344 | - | - |
| *max_sig22_top* | 0.0030 | **0.0016** | 0.0022 | 0.0030 | - | - |
| **total_error** | 0.0673 | **0.0027** | 0.0337 | 0.0154 | - | - |
| **2D_MultiScHypEl** | | | | | | |
| **u1** | 0.0400 | - | 0.0350 | **0.0140** | - | - |
| **u2** | 0.0444 | - | 0.0356 | **0.0164** | - | - |
| **P11** | 0.0383 | - | 0.0611 | **0.0185** | - | - |
| **P12** | 0.0670 | - | 0.1016 | **0.0316** | - | - |
| **P22** | 0.0383 | - | 0.0614 | **0.0189** | - | - |
| **P21** | 0.0663 | - | 0.1005 | **0.0311** | - | - |
| **psi** | 0.0443 | - | 0.0580 | **0.0239** | - | - |
| *effective_energy* | 0.0111 | - | **0.0108** | 0.0313 | - | - |
| **total_error** | 0.0437 | - | 0.0580 | **0.0232** | - | - |
| **2D_ElPlDynamics** | | | | | | |
| **U_x** | - | - | - | - | **0.0025** | - |
| **U_y** | - | - | - | - | **0.0291** | - |
| **total_error** | - | - | - | - | **0.0158** | - |
| **Rotor37** | | | | | | |
| **Density** | 0.0114 | **0.0039** | 0.0370 | 0.0055 | - | - |
| **Pressure** | 0.0114 | **0.0039** | 0.0366 | 0.0053 | - | - |
| **Temperature** | 0.0024 | **0.0009** | 0.0074 | 0.0012 | - | - |
| *Massflow* | 0.0061 | **0.0007** | 0.0058 | 0.0028 | - | - |
| *Compression_ratio* | 0.0060 | **0.0007** | 0.0055 | 0.0028 | - | - |
| *Efficiency* | 0.0071 | **0.0009** | 0.0067 | 0.0019 | - | - |
| **total_error** | 0.0074 | **0.0019** | 0.0165 | 0.0033 | - | - |
| **2D_profile** | | | | | | |
| **Mach** | 0.0604 | **0.0514** | 0.0699 | - | - | - |
| **Pressure** | 0.0466 | **0.0335** | 0.0430 | - | - | - |
| **Velocity-x** | 0.0735 | **0.0585** | 0.0854 | - | - | - |
| **Velocity-y** | 0.0566 | **0.0483** | 0.0570 | - | - | - |
| **total_error** | 0.0593 | **0.0480** | 0.0638 | - | - | - |
| **VKI-LS59** | | | | | | |
| **nut** | 0.1656 | 0.0822 | 0.1489 | 0.0641 | - | **0.0259** |
| **mach** | 0.0451 | 0.0309 | 0.0643 | 0.0245 | - | **0.0112** |
| *Q* | 0.0716 | **0.0023** | 0.0228 | 0.0076 | - | 0.0052 |
| *power* | 0.0403 | **0.0057** | 0.0168 | 0.0108 | - | 0.0077 |
| *Pr* | 0.0068 | 0.0026 | 0.0042 | 0.0050 | - | **0.0018** |
| *Tr* | 0.0001 | **0.0000** | 0.0001 | **0.0000** | - | **0.0000** |
| *eth_is* | 0.1912 | 0.1224 | 0.1311 | 0.1732 | - | **0.0453** |
| *angle_out* | 0.0263 | 0.0033 | 0.0061 | 0.0040 | - | **0.0023** |
| **total_error** | 0.0684 | 0.0312 | 0.0493 | 0.0362 | - | **0.0124** |

**Table 4:** RRMSE and total_error on PLAID benchmarks, best on each line is **bold**, second best is underlined.

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

# A    Details on the ML models used in the benchmark

We briefly present the main competing methods that we used for the benchmark. We also highlight some practical details about their implementation. Readers are encouraged to refer directly to the papers introducing the methods for further information.

## A.1    MGN

### A.1.1    Method

MeshGraphNet (MGN) [25], introduced by T. Pfaff et al., is a framework designed for learning mesh-based simulations using graph neural networks. The model is capable of being trained to simulate dynamic solutions by passing messages over a meshed domain, predicting acceleration at each mesh node at a given time step. This prediction allows for the calculation of the output field at the next time step through forward integration. Specifically, MGN is trained using one-step supervision and can be applied iteratively to generate long trajectories during inference. The architecture of MeshGraphNet is composed of encoding, processing, and decoding steps. In this work, MGN has been adapted to predict steady-state fields.

We utilize the following features as input (see Figure 5 for the workflow diagram):

- the distance of each node to the boundary,
- the type of node,
- the coordinates of the node.

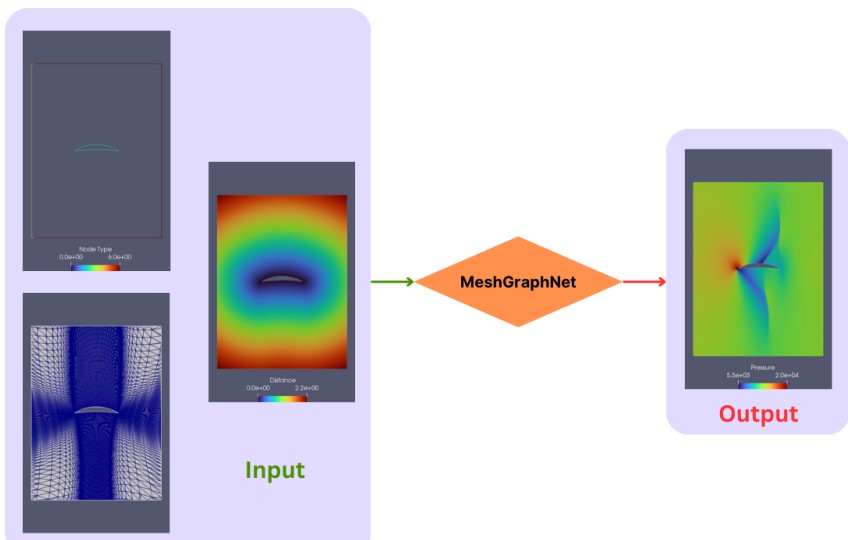

**Figure 5:** Illustration of MGN workflow to predict steady-state pressure field of a sample from the `2D_profile` dataset.

### A.1.2    Experiments

In this section, we provide a summary of the experiments conducted on various datasets.

For all datasets, we trained two separate models: one focused on field predictions and the other on scalar predictions. For scalar outputs, a readout layer taken from [89] is added to the model. Except for the `2D_profile` dataset, we only required a single model since it does not include scalar prediction tasks.

The LeakyReLU is chosen as the activation function, and all models are trained for 1000 epochs.

The input node features consist of those introduced in the previous section, combined with input scalars if they exist. Given two node coordinates $x_i$ and $x_j$, the calculation for edge features is based on $\exp(-||x_i - x_j||_2^2/(2h^2))$, where $h$ epresents the median value of the edge lengths within the mesh.

The rest of architecture details and training information are outlined in Table 5 and Table 6.

| Dataset | Message Passing Steps | Latent Size | Nbe epochs | Batch size | Training Time | Hardware |
|---|---|---|---|---|---|---|
| Tensile2d | 10 | 16 | 1000 | 1 | 3h46min | $1 \times$ A100 |
| 2D_MultiScHypEl | 10 | 32 | 1000 | 1 | 5h54min | $1 \times$ A100 |
| Rotor37 | 10 | 64 | 1000 | 1 | 19h24min | $1 \times$ A100 |
| 2D_profile | 10 | 128 | 1000 | 1 | 17h27min | $1 \times$ A100 |
| VKI-LS59 | 10 | 64 | 1000 | 1 | 16h32min | $1 \times$ A100 |

**Table 5:** Field MGN: Architecture details and training statistics across datasets.

| Dataset | Message Passing Steps | Latent Size | Nbe epochs | Batch size | Training Time | Hardware |
|---|---|---|---|---|---|---|
| Tensile2d | 10 | 32 | 1000 | 1 | 4h6min | $1 \times$ A100 |
| 2D_MultiScHypEl | 10 | 16 | 1000 | 1 | 6h | $1 \times$ A100 |
| Rotor37 | 10 | 16 | 1000 | 1 | 10h | $1 \times$ A100 |
| VKI-LS59 | 10 | 16 | 1000 | 1 | 9h13min | $1 \times$ A100 |

**Table 6:** Scalar MGN: Architecture details and training statistics across datasets.

## A.2   MMGP

### A.2.1   Method

We refer the reader to [19] for a complete presentation of the Mesh Morphing Gaussian Process (MMGP) method. MMGP combines four main ingredients: (i) mesh morphing, (ii) finite element interpolation, (iii) dimensionality reduction, and (iv) Gaussian process regression. Together, these enable learning mappings between geometries and solution fields for PDEs, even when the input geometry is provided as non-parametrized meshes.

An overview of the workflow is illustrated in Figure 6, which should be read from left to right. On the left are sample-specific input geometries; on the right are the corresponding solution fields defined on these geometries.

Since input meshes are not parametrized, they must first be embedded into a learnable space. MMGP does this by interpreting mesh vertex coordinates as continuous fields (e.g., the $x$-coordinate field shown in the left column of Figure 6, exhibiting vertical iso-lines). Each mesh is then deterministically morphed onto a reference geometry—the unit disk in this 2D example, but it can be one of the training samples shape. Next, each sample morphed coordinate fields are projected onto a common mesh of the reference geometry via finite element interpolation. This ensures all samples share a consistent discretization, making them compatible with standard dimensionality reduction techniques like PCA. The result is a compact, fixed-size representation of the geometry. When scalar inputs are present, they can be concatenated to the reduced vector.

A similar procedure is applied to the output fields: morphing onto the reference geometry, projection onto the common mesh, and PCA compression yield low-dimensional field representations aligned with the geometric embeddings.

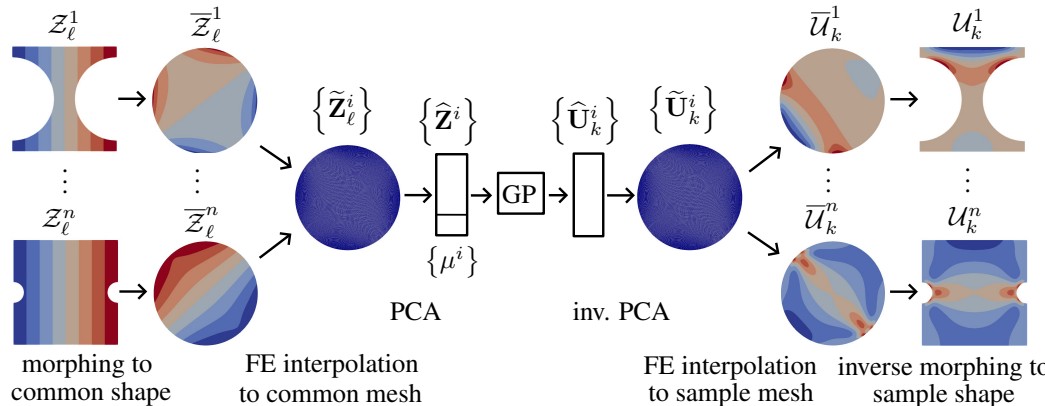

**Figure 6:** Illustration of the MMGP inference workflow for the prediction of an output field of interest [19].

These deterministic preprocessing steps transform the original complex problem—mapping between high-dimensional and irregularly discretized fields—into a standard regression task between low-dimensional vectors. This enables the use of classical regression models; we adopt Gaussian process regression due to its robustness, accuracy, and built-in uncertainty quantification.

MMGP offers several practical advantages: it handles large meshes efficiently, produces interpretable models, and delivers high accuracy in our experiments, with uncertainty estimates. In industrial design applications, where data can lie on low-dimensional manifolds, small models like MMGP can be especially effective—provided that the reparametrization (or embedding) is constructed appropriately, here with the morphing.

The main limitations of MMGP are tied to the morphing step, which currently requires problem-specific setup, and the fact that morphing and interpolation must still be performed at inference time. These challenges are addressed in recent works [20, 90], which introduce automatic alignment and online-efficient morphing strategies. Further improvements are proposed in [21], where optimization techniques are used to generate morphings that maximize PCA compression.

All mesh and field operations are implemented using the Muscat library [67, 68]. An upcoming release will include a GPU-accelerated finite element interpolation routine, significantly improving inference latency.

Additional improvement of MMGP are possible, by replacing the linear decoder of the PCA by a non-linear one that accounts for high-order interactions among the selected POD modes and includes a rotation of the POD basis and a polynomial correction, as proposed in [91].

Physics-based models compatible with the morphing, finite element interpolation and dimensionality reduction of MMGP have been proposed. The physics equation can be efficiently assembled and solved on the low-dimension space spanned by the PCA modes obtained after morphing, instead of using data-driven low-dimensional models. In [92], a hyper-reduced least-square Petrov-Galerkin scheme is used to reduced the Navier-Stokes equations, with morphing. While much more complicated to utilize, we expect such methods to greatly improve the accuracy, with a moderate additional computation cost.

### A.2.2 Experiments

Hyperparameters and training statistics for the MMGP experiments are listed in Table 7. We first mention that MMGP has not been applied to the `2D_ElPlDynamics` and `2D_MultiScHypEl` datasets, since the method is yet to be extended to variable topology settings.

We notice that `Rotor37` and `VKI-LS59` do not require morphing, since the samples' meshes have the same number of nodes. In `Tensile2d` and `2D_profile`, systematic morphing strategies to align the shapes are sufficient, with respectively Tutte barycentric embedding [19, Ann B] and elasticity-based automatic morphing [20].

Since the `VKI-LS59` dataset exhibits discontinuities due to the presence of shock waves, a non-linear decoder [91] was employed to reconstruct the fields of interest. For the compression of the `mach`

fields, 5 POD modes and a polynomial order of 3 were used, while 40 POD modes were retained for the compression of the `nut` fields. Since polynomial decoders are prone to overfitting, the number of modes and the polynomial order were selected through a $k$-fold cross-validation procedure on the training set.

Since the solution fields of `2D_profile` and `VKI-LS59` feature complex structures (e.g. shocks of variable position), we expect the involved optimal morphing strategy from [21] to significantly improve the results of MMGP on these cases.

| Dataset | Morphing | PCA modes (shape) | PCA modes (field) | GP kernel | Training time | Hardware |
|---------|----------|-------------------|-------------------|-----------|---------------|----------|
| Tensile2d | Tutte [19, Ann B] | 8 | 8 | Matérn 5/2 | 13min02s | 128 cores |
| Rotor37 | None | 32 | 64 | Matérn 5/2 | 6min13s | 128 cores |
| 2D_profile | Elasticity [20] | 16 | 32 | RBF | 18min32s** | 12 cores |
| VKI-LS59 | None | 13 | 5-3/40-1* | Matérn 5/2 | 4min13s | 64 cores |

**Table 7:** Hyperparameters and training statistics for the MMGP experiments (on an AMD EPYC 9534 CPU). Training times include all preprocessing (morphing, finite element interpolation and dimensionality reduction), in addition to the training of the Gaussian processes. *For `VKI-LS59`, X-Y stands for the number of modes and polynomial order of the decoder for the `mach` and `nut` fields respectively. **Not including morphing time (which takes approximately 10min on 300 cores).

### A.3 Vi-Transformer and Augur

#### A.3.1 Method

**Transformers for long context range regression.** The natural way of dealing with mesh-based regression problems is to use GNN models which rely on message-passing. Although these are great at capturing information locally, they struggle to retrieve it at long distances. Indeed, the smallest number of GNN layers needed to have a receptive field that covers the whole graph is half the diameter of the graph. This becomes computationally impractical in the context of large simulation meshes. This behavior is analogous to Convolutional Neural Networks (CNNs) in Computer Vision (CV) where long-range dependencies are only captured at the deeper levels of the network. One way of alleviating this is to consider transformer architectures, which compute similarities between all the input tokens simultaneously thanks to the attention mechanism, thus removing the need to have infeasibly deep networks.

**Transformers on very large data.** One of the main challenges of transformers in this case is to handle the large size of the meshes (in the order of tens of thousands of points per mesh, and up to millions with practical industrial problems). Currently, the computational bottleneck of transformers is a widely considered subject: given $N$ tokens of dimension $D$, the critical issue of self attention is that one needs $N^2 \times \mathcal{D}$ operations where $\mathcal{D} \approx D$ is the size of the embedding of each token, and $N^2$ is the cost to compute the Gram matrix of the $N$ tokens (this computation cost is also a memory one as storing the matrix requires also $N^2\mathcal{D}$ numbers).

Many papers have focused on the possibility to linearize the cost of self-attention, for example:

- [93] introduces Reformer which considers the formulation of the attention mechanism : $\mathrm{softmax}\left(\frac{QK}{\sqrt{D}}\right)$ with the key and query matrices (respectively $K$ and $Q$), capitalizing on the fact that for a given query $Q_i$, only the keys which provide high dot products with $Q_i$ will have a significant impact on the value of $\mathrm{softmax}\left(\frac{Q_i K^T}{\sqrt{D}}\right)$. Therefore, Reformer makes use of locality-sensitive hashing for only computing the $Q_i K_j^T$ products with the $p$ keys that are closest to a query, where $p \in \mathbb{N}$ is a chosen hyperparameter, efficiently linearizing the self attention.

- [94] introduces Linformer. Coarsely, Linformer relies on the Nyström approximation to approximate the Gram matrix of self attention. Precisely, while the Nyström approximation replaces an $n \times n$ symmetric matrix $A$ by $UU^T$ where $U$ is only $n \times k$ containing the eigenvec-

tors of largest eigenvalue, Linformer offers to learn $E, F$ such that $\text{softmax}\left(\frac{QK}{\sqrt{D}}\right) \approx EF^T$. This also offers a linear approximation of the self attention computation.

This has also been tackled in CV tasks [95], where self-attention is not applied on pixels directly but on pixel-patches that aggregate pixel neighborhoods into tokens, thus drastically reducing the self-attention's input sequence length.

**Transformers for large scale point-wise regression.**  The most used transformer architectures are in one of two categories. The auto-regressive sequence-to-sequence transformers, mostly used in Natural Language Processing (NLP) for text generation, and the sequence-to-class ones which are used both in NLP, as in sentiment analysis [96], spam detection [97], long document classification [98], and CV with image classification [95].

Both are quite different from the point-wise regression objective of the PLAID benchmarks. Indeed, the first method generates new token sequences of arbitrary lengths, while the second only makes use of transformer encoders with neural network heads to obtain a probability distribution on a set of classes.

Some work has been conducted in order to tackle regression problems with transformers:

- Segformer [99] addresses this in the case of image segmentation; it uses a multiscale U-type transformer to sequentially downscale the input image, and uses a multiscale MLP head to decode these downscaled states into the output segmentation mask.
- Point Transformer [100] also uses a U-style encoder-decoder architecture, this time on 3D point-cloud data for both segmentation and classification.
- TransCFD [101] tackles airfoil surrogate CFD modeling by using a decoder-only architecture from a latent embedding of the input geometry. It relies on structured regular grids (images) of the inputs, and not arbitrary mesh discretizations.
- Point Transformer V3 [102] groups points together and computes attention scores within these groups. Local and long-distance information are captured through different serializations of the input mesh.

Both Segformer and TransCFD make use of the regular nature of their data to precisely decode (and/or encode) the output (and/or input) fields. Point Transformers, on the other hand, handle unstructured point-cloud data. Although these methods fit the nature of the PLAID benchmark, we propose lighter methods that stick more closely to the classical transformer model.

**Vi-Transformer for mesh field regression.**  The chosen approach relies on a transformer encoder architecture and is analogous to Vision Transformers (Vi-Transformer). Rather than considering each node of the mesh as a token by its own, the encoder takes as input tokenized point-cloud patches. Local information is kept within the patches while long-range information is retrieved through the transformer's mapping, which compares all token pairs together. The general architecture of the Vi-Transformer is depicted in Figure 7.

**Augur Transformer model.**  Augur has developed Transformer models specifically designed for numerical simulations. These models share fundamental architectural similarities with Vision Transformers (ViT), where the computational mesh is decomposed into patches. Each patch is embedded into a latent space, resulting in the input tokens for the Transformer architecture. This approach enables information exchange between local patches across long spatial distances, similar to how ViTs process image data.

The key innovation in Augur's approach lies in the decoding mechanism, addressing a critical question: how to properly reconstruct the output field from the processed sequence of tokens? In traditional ViT architectures, direct reconstruction from individually processed tokens can result in discontinuities at patch interfaces due to insufficient global context integration. Augur models overcome this limitation by incorporating a global information vector that aggregates data from all tokens. The decoder then uses a combination of point-specific information, processed local features, and global context to produce a more robust and consistent output field. Furthermore, unlike ViTs, Augur models do not treat scalar predictions as constant fields but instead derive them directly from

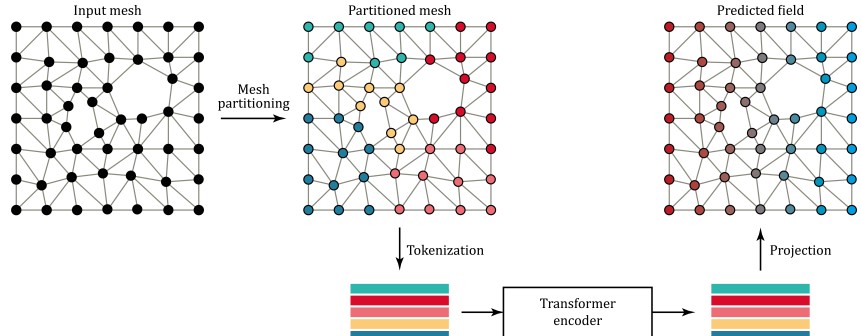

**Figure 7:** Vi-Transformer architecture. Input meshes are partitioned using the Metis domain decomposition algorithm [103]. Each such sub-domain is then tokenized before passing through the transformer encoder. In the end, each token is decoded into its domain's corresponding fields. Input scalars are embedded during the tokenization procedure while output scalars are estimated as uniform fields.

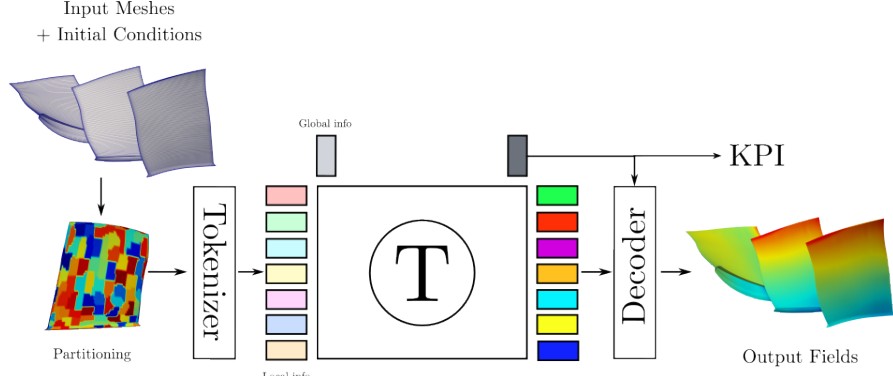

**Figure 8:** Augur Transformer architecture: Input meshes are partitioned using the Metis domain decomposition algorithm. Each subdomain is then tokenized before being passed through the Transformer. An additional global tensor is added to the Transformer to gather global information. Output fields are reconstructed using a decoder that leverages both local and global information. Output scalars (KPIs) are predicted directly from the global tensor.

the global information vector, enhancing prediction accuracy. The general architecture of the Augur model is depicted in Figure 8.

### A.3.2 Experiments

Both the Vi-Transformer and Augur models rely on a relatively small number of hyperparameters. These include the patch size (i.e., the number of nodes per patch), the latent dimension onto which the aggregated patches are projected, and the Transformer encoder hyperparameters, such as the number of heads, the number of transformer encoder layers and the dimension of the feedforward layer. Table 8 details the hyperparameters for the Vi-Transformer, while Table 9 outlines those for the Augur model.

### A.4 DAFNO

DAFNO belongs to the Operator Approximator class of architectures, i.e. it builds mappings between two function spaces. The use of the Fast Fourier Transform (FFT) within the different layers leads to the sampling of the input function on a regular grid, thus falling back to a finite dimension space. This architecture has the advantage of learning transformation in the frequency domain which provides a significant advantage compared to CNNs on several physical problems.

| Dataset | Patch size | Latent dimension | Feedforward dimension | Nb encoder layers | Training time | Hardware |
|---------|-----------|------------------|----------------------|-------------------|--------------|----------|
| Tensile2d | 50 | 6400 | 2048 | 5 | 3h18 | $3 \times$ A30 |
| 2D_MultiScHypEl | 10 | 512 | 256 | 5 | 1h56 | $3 \times$ A30 |
| Rotor37 | 100 | 256 | 256 | 10 | 33min | $3 \times$ A30 |
| 2D_profile | 50 | 1024 | 1024 | 5 | 36min | $3 \times$ A30 |
| VKI-LS59 | 50 | 6400 | 2048 | 5 | 1h27 | $3 \times$ A30 |

**Table 8:** (Vi-Transformer) Hyperparameters and training statistics for the Vi-Transformer experiments. Training times include all preprocessing (domain decomposition, tokenization), in addition to the training of the model itself. The number of attention heads is kept at 16 for all experiments.

| Dataset | Patch size | Latent dimension | Feedforward dimension | Nb encoder layers | Training time | Hardware |
|---------|-----------|------------------|----------------------|-------------------|--------------|----------|
| Tensile2d | 16 | 512 | 2048 | 8 | 1h11 | $1 \times$ RTX 2080Ti |
| 2D_MultiScHypEl | 4 | 128 | 512 | 8 | 7h48 | $1 \times$ RTX 2080Ti |
| Rotor37 | 32 | 256 | 1024 | 8 | 2h30 | $1 \times$ RTX 2080Ti |
| VKI-LS59 | 64 | 512 | 2048 | 4 | 2h15 | $1 \times$ RTX 2080Ti |

**Table 9:** (Augur) Hyperparameters and training statistics for the Augur experiments. Training times include all preprocessing (domain decomposition, tokenization), in addition to the training of the model itself.

### A.4.1 Method

The DAFNO model deals separately with the input fields and the geometry of the problem [86]: let $u : \mathbb{R}^2 \to \mathbb{R}^k$ be our input fields and $\chi_\Omega : \mathbb{R}^2 \to \{0, 1\}$ be the characteristic function of the domain $\Omega$. Let $W \in \mathbb{R}^{k \times k}$, $W^* \in \mathbb{R}^{k \times k}$, $c \in \mathbb{R}^k$ be the learnable parameters, let $\sigma : \mathbb{R} \to \mathbb{R}$ be a scalar non-linear function (sigmoid, ReLU, or tanH) to be applied elementwise. A layer of the DAFNO architecture is defined by the following operator:

$$\mathcal{J}[u](x) = \sigma \left( Wu(x) + c + \mathcal{F}^{-1} \left[ W^* \mathcal{F} \left[ (u(x) - u(\cdot)) \chi_\Omega(\cdot) \chi_\Omega(x) \right] \right] (x) \right) \tag{1}$$

$$= \sigma \left( Wu(x) + c + \chi_\Omega(x) \mathcal{I} \left[ \chi_\Omega(\cdot) u(\cdot) \right] (x) - u(x) \mathcal{I} \left[ \chi_\Omega(\cdot) \right] (x) \right), \tag{2}$$

where $\mathcal{I}[f](x) = \mathcal{F}^{-1} \left[ W^* \mathcal{F}[u](\cdot) \right] (x)$, with $\mathcal{F}$ denoting the FFT operator. Equation (1) shows the interest of using the DAFNO architecture: the FFT in operator only considers values inside the domain $\Omega$. Moreover the FFT is computed over the local variation of the input field rather than the input field itself ($u(x) - u(.)$ instead of $u(.)$) making the layer, by design, seek features within local variations. The DAFNO network ends up being a composition of one or multiple of such layers. The mask $\chi_\Omega$ is used at each layer unaltered to make sure that no noise outside the domain may perturb the prediction.

FNO models and variant can only predict on regular grids (this is due to the use of the FFT). This is a common constraint shared with some neural networks such as CNNs. This means that, in order to predict on an unstructured mesh, a preprocessing and postprocessing of the fields are needed. The preprocessing consists in a projection of the original mesh to a regular grid where the FNO is able produce a prediction. Then, a postprocessing projecting back from the regular grid to the original mesh needs to be performed to compare the prediction to reference fields. The projection operations were performed using Muscat [68, 67].

### A.4.2 Experiments

The DAFNO architecture can build transient predictions on various geometries and topology, the only dataset introduced in Section 4 that meets these three characteristics is the 2D_ElPlDynamics dataset.

**Training procedure.** The training was performed in a autoregressive manner: given the input fields at time $t$, the model has to predict the fields at time $t + dt$ very much like an explicit solver would do.

Once trained, one may build the whole transient field prediction by applying the model recursively on the initial conditions. A key choice involves selecting inputs that are informative enough for the model to accurately predict the system dynamics.

On top of the fields provided by the dataset (U_x and U_y) we added two coordinate fields (one for $x$ and one for $y$) and we computed a fifth input: a smoothed mask $\chi_\Omega^{\text{smooth}}$ as suggested by the original DAFNO paper [86] along with being a drop in replacement of $\chi_\Omega$ in the DAFNO layers. This smooth quantity is richer than its discontinuous counterpart since it provides insight on how close we are from the border of the geometry. We are summarizing the input/output quantities in Table 10.

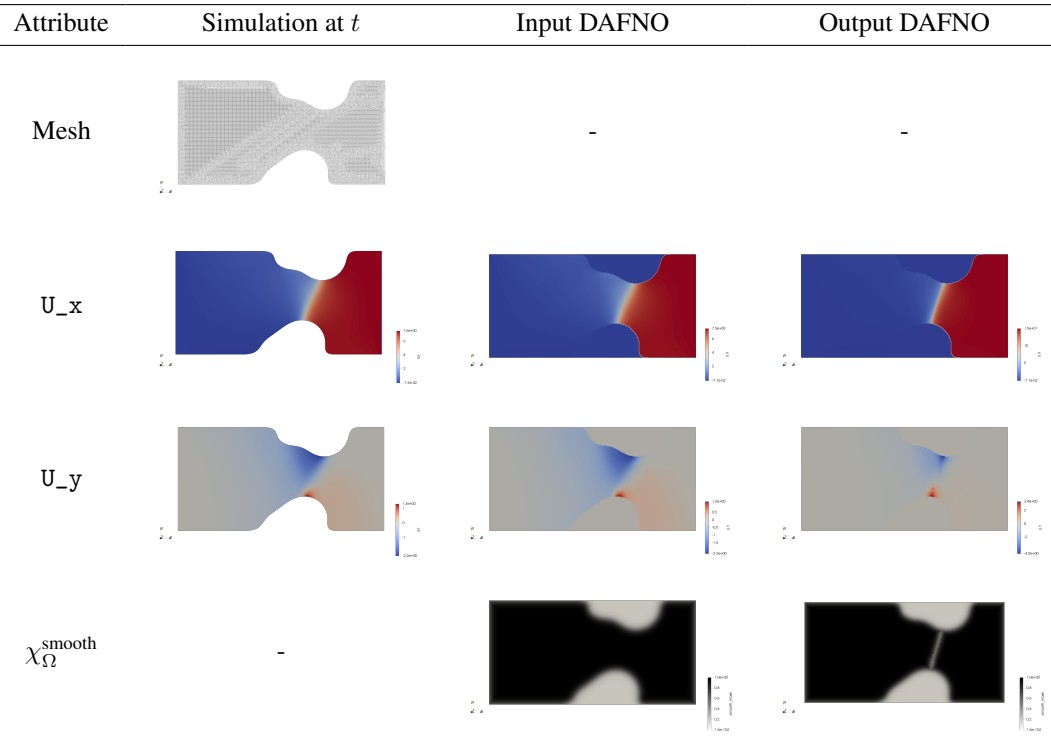

| Attribute | Simulation at $t$ | Input DAFNO | Output DAFNO |
|-----------|-------------------|-------------|--------------|
| Mesh | | - | - |
| U_x | | | |
| U_y | | | |
| $\chi_\Omega^{\text{smooth}}$ | - | | |

**Table 10:** Features throughout the learning process, "-" means the field is not available/used at the given stage. The simulation at $t$ (column 1) can be projected to a regular grid (column 2). The regular fields along with coordinates fields $x$ and $y$ make input features for the DAFNO model which in turns predicts the fields at $t + dt$ (column 3).

The training was parallelized on 40 GPUs (A100) and lasted 6 hours. Inference and thus testing can be performed on a single GPU to compute the metrics presented in Table 4.

**Model and training parametrization.** We summarize the model parametrization in Table 11 and training procedure in Table 12.

| Model parameters | Layer count | Channel hidden layers | Padding | Fourier modes | Activation Function |
|------------------|-------------|-----------------------|---------|---------------|---------------------|
| Value | 8 | 64 | 8 | $20 \times 20$ | GELU |

**Table 11:** DAFNO: parametrization of the model.

## A.5 MARIO

Modulated Aerodynamic Resolution-Invariant Operator (MARIO) is a deep learning model designed to approximate the solution operator of a partial differential equation (PDE) [87], involving geometric variability. It leverages Conditional Neural Fields (or Implicit Neural Representations) to learn

| Training parameters | Epochs | Optimizer | Learning rate | Batch size | Loss |
|---|---|---|---|---|---|
| Value | 1800 | Adam | 0.0003 | 60 | Pixel-wise $L_2$ |

**Table 12:** DAFNO: parametrization of the training.

the mapping between spatial coordinates from a mesh, geometric information (e.g., via the signed distance function, SDF), inflow conditions, and the resulting physical field. Unlike mesh-based methods, INRs represent continuous fields through neural network parameterizations, enabling resolution-independent predictions and flexible evaluation. MARIO extends this approach to handle multiple geometries and operating conditions through a conditioning mechanism.

### A.5.1 Method

**Modulated INR architecture.** MARIO implements a conditional neural field approach where a single neural network architecture can represent multiple distinct signals through a conditioning mechanism. The conditioning variable $z = [\mu_{\text{geom}}, \mu]$ encodes both geometric parameterization $\mu_{\text{geom}}$ and operating conditions $\mu$ (e.g., angle of attack, Mach number, Reynolds number).

The main network is a multilayer perceptron (MLP) where the layer outputs are modulated by sample-specific vectors:

$$f_{\theta,\phi}(x) = W_L(\eta_{L-1} \circ \eta_{L-2} \circ \cdots \circ \eta_1 \circ \gamma(x)) + b_L \tag{3}$$
$$\eta_l(\cdot) = \text{ReLU}(W_l(\cdot) + b_l + \phi_l(z)) \tag{4}$$

where $\phi_l(z) = [h_\psi(z)]_l \in \mathbb{R}^{d_l}$ are layer-specific modulation vectors obtained from the hypernetwork $h_\psi$ that processes the conditioning variable $z$. The main network parameters $\theta$ are shared for all samples and consist of the weights and biases matrices $W_l, b_l$. In MARIO, an explicit shape encoding $\mu_{\text{geom}}$ is used as input of the architecture to properly model geometric variability. In many real-world applications, a geometric parameterization is not available or insufficient to capture complex shapes. Therefore, a learning mechanism to obtain compact geometric representations from the SDF fields is adopted. These encoding process leverages a separate Neural Field encoder, that maps input coordinates to output SDF values, while fitting latent shape representations.

**Geometry encoding mechanism.** For each geometry's signed distance function (SDF), a meta-learning optimization procedure based on CAVIA [104] adapts a shared neural network $f_{\theta_{in},\phi_{in}}$ to represent different shapes. Given the shared network parameters $\theta_{in}$ and hypernetwork parameters $\psi$, the latent representation $\mu_{geom} = z_{in}^{(K)}$ for geometry $i$ is obtained by solving:

$$z_{in}^{(0)} = 0 \tag{5}$$
$$z_{in}^{(k+1)} = z_{in}^{(k)} - \alpha \nabla_{z_{in}^{(k)}} \mathcal{L}_{in}(f_{\theta_{in},\phi_{in}}(x), sdf_i), \quad \text{for } 0 \leq k \leq K - 1 \tag{6}$$

where $\phi_{in} = h_\psi(z_{in}^{(k)})$, $\alpha$ is the inner loop learning rate, and $K$ is the number of optimization steps (typically set to 3). The loss $\mathcal{L}_{in}$ measures the reconstruction error between the true SDF field and its prediction over a sampling grid defined on the input domain.

This optimization process, illustrated in Figure 9, yields a compact latent code $\mu_{\text{geom}} = z_{in}^{(K)}$ that captures the essential geometric features.

**Fourier feature encoding.** To address the spectral bias inherent in neural networks, MARIO employs Fourier feature encoding for the input coordinates:

$$\gamma(x) = [\cos(2\pi \mathbf{B}x), \sin(2\pi \mathbf{B}x)] \tag{7}$$

where $\mathbf{B} \in \mathbb{R}^{m \times d}$ contains frequency vectors sampled from a Gaussian distribution $\mathcal{N}(0, \sigma)$. This encoding enables the network to better capture high-frequency details in the output fields.

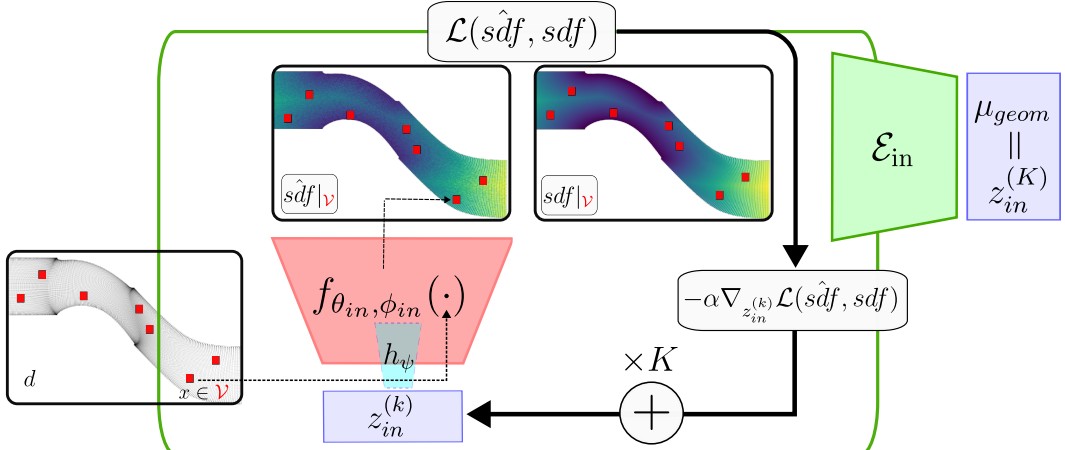

**Figure 9:** MARIO geometry encoding process.

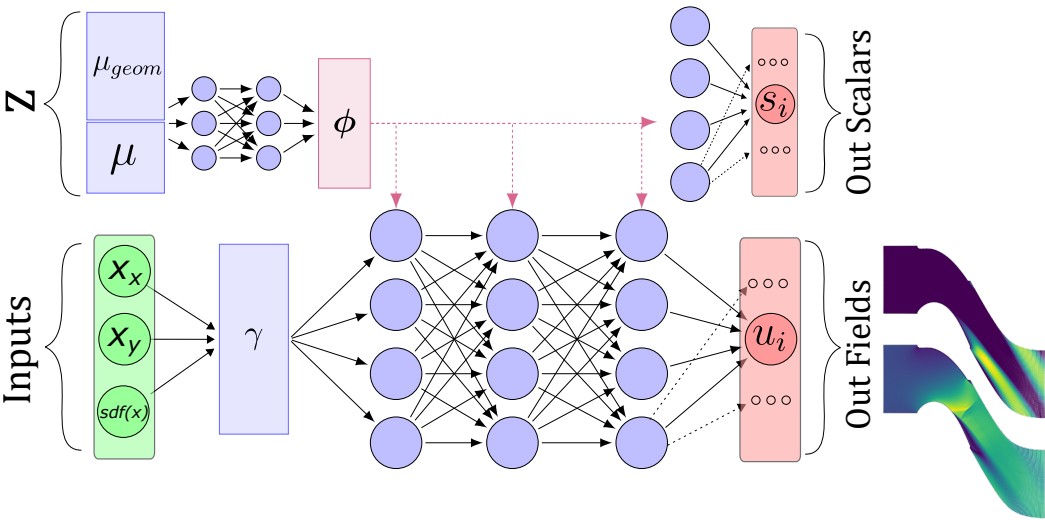

**Figure 10:** MARIO architecture.

**Scalar output prediction.** In addition to predicting coordinate-dependent fields, MARIO can also predict global scalar quantities for each sample. Since these scalar outputs are global properties of the solution (e.g., power coefficients, efficiency metrics), they depend only on the sample-specific information encoded in the modulation vectors. The scalar prediction is therefore implemented as:

$$s = W_s \cdot \phi_{agg} + b_s \tag{8}$$

where $\phi_{agg}$ represents an aggregation of the modulation vectors produced by the hypernetwork. This single-layer transformation efficiently leverages the already learned sample representation without requiring additional feature extraction.

The architecture of MARIO is illustration in Figure 10.

**Training procedure.** MARIO is trained using a weighted loss function that balances field prediction accuracy and scalar output accuracy:

$$\mathcal{L} = \alpha \cdot \mathcal{L}_{\text{field}} + (1 - \alpha) \cdot \mathcal{L}_{\text{scalar}} \tag{9}$$

where $\alpha \in [0, 1]$ is a weighting parameter. The field loss $\mathcal{L}_{\text{field}}$ is computed as the mean squared error between predicted and target fields across spatial locations, while the scalar loss $\mathcal{L}_{\text{scalar}}$ is the mean squared error of the global quantities.

**Key advantages.** MARIO presents three major benefits: (i) it is resolution-invariant and can be evaluated at arbitrary spatial locations; (ii) it overcomes spectral bias through multiscale Fourier encodings; and (iii) it adapts to geometry-specific variations via bias modulation using the auxiliary network $h_\psi$.

### A.5.2 Experiment

**Model and training parametrization.** The model parametrization and training procedure are provided respectively in Tables 13 and 14.

| Model param. | Geom. Hypernet. depth | Geom. Hypernet. width | Geom. latent dim | Hypernet. depth | Hypernet. width | INR depth | INR width | Nb of frequencies |
|---|---|---|---|---|---|---|---|---|
| Value | 1 | 128 | 16 | 3 | 256 | 6 | 256 | 64 |

**Table 13:** MARIO: parametrization of the model.

| Training param. | Epochs | Optimizer | Learning rate | Batch size | Training time | Training hardware | Loss ($\alpha = 0.8$) |
|---|---|---|---|---|---|---|---|
| Value | 2000 | AdamW | 0.001 | 4 | 30h | $1 \times$ A100 | MSE |

**Table 14:** MARIO: parametrization of the training.

We notice that MARIO is significantly longer to train than the other tested models.

## B  Additional details on PLAID

We illustrate further the capabilities of PLAID by providing some additional commands to retrieve information from our datasets directly from Hugging Face.

### B.1  `Tensile2d`

`Tensile2d` is a simple dataset, for which standard and simple PLAID commands are sufficient to retrieve the data:

```python
from datasets import load_dataset
from plaid.containers.sample import Sample
import pickle

# Load the dataset
hf_dataset = load_dataset("PLAID-datasets/Tensile2d", split="all_samples")

# Get split ids
ids_train = hf_dataset.description["split"]["train_500"]

# Get inputs/outputs names
in_scalars_names = hf_dataset.description["in_scalars_names"]
out_fields_names = hf_dataset.description["out_fields_names"]

# Get samples
sample = Sample.model_validate(pickle.loads(hf_dataset[ids_train[0]]["sample"]))

# Examples of data retrievals
nodes = sample.get_nodes()
elements = sample.get_elements()
```

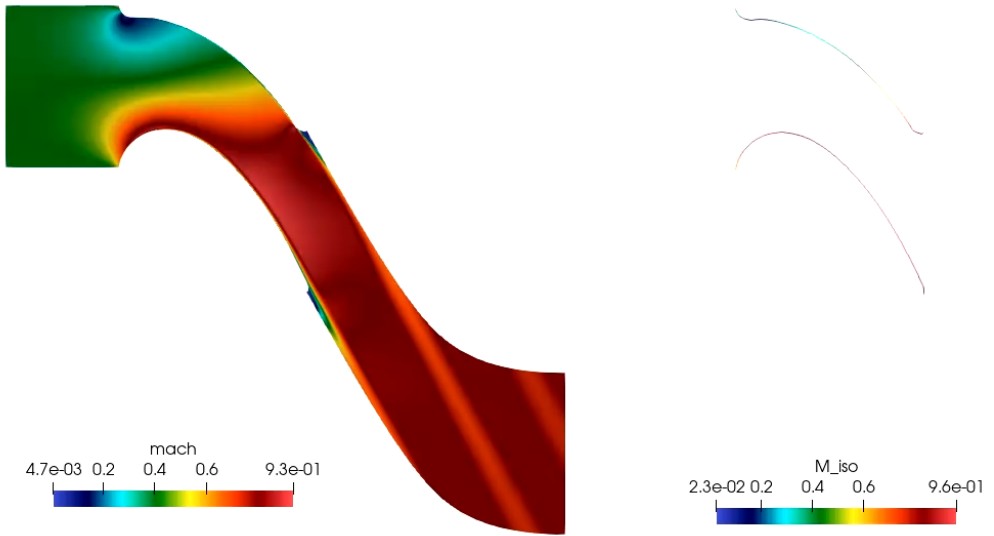

**Figure 11:** Illustration of the first sample in the train split of `VKI-LS59`: (left) fluid domain, (right) blade surface domain.

```
21   nodal_tags = sample.get_nodal_tags()
22
23   for sn in ["P", "p1", "p2", "p3", "p4", "p5"]:
24       scalar = sample.get_scalar(sn)
25
26   # outputs
27   for fn in ["U1", "U2", "q", "sig11", "sig22", "sig12"]:
28       field = sample.get_field(fn)
29
30   for sn in ["max_von_mises", "max_q", "max_U2_top", "max_sig22_top"]:
31       scalar = sample.get_scalar(sn)
32
```

834   The geometrical support in PLAID samples can be easily converted to Muscat meshes:

```
1   from Muscat.Bridges import CGNSBridge
2   CGNS_tree = sample.get_mesh()
3   mesh = CGNSBridge.CGNSToMesh(CGNS_tree)
```

835   **B.2** `VKI-LS59`

836   `VKI-LS59` also contains stationary configurations, meaning only one time step per sample, but
837   features a complex geometrical setting, with a 2D fluid domain and a 1D blade surface domain, see
838   Figure 11.

839   The fluid domain contains 2D elements in a 2D ambient space, hence is contained in the CGNS base
840   called "Base_2_2". For the blade surface domain, we have 1D elements in a 2D ambient space: the
841   CGNS base is then "Base_1_2". The corresponding data are retrieved as follows:

```
1   from datasets import load_dataset
2   from plaid.containers.sample import Sample
3   import pickle
4
5   # Load the first sample of the train split
6   hf_dataset = load_dataset("PLAID-datasets/VKI-LS59", split="all_samples")
```

```
7    ids_train = hf_dataset.description["split"]["train"]
8    sample = Sample.model_validate(pickle.loads(hf_dataset[ids_train[0]]["sample"]))
9
10   # Examples of data retrievals
11   for fn in ["sdf", "ro", "rou", "rov", "roe", "nut", "mach"]:
12       field = sample.get_field(fn, base_name="Base_2_2")
13   M_iso = sample.get_field("M_iso", base_name="Base_1_2")
14   for sn in sample.get_scalar_names():
15       scalar = sample.get_scalar(sn)
16
17   nodes_fluid = sample.get_nodes(base_name="Base_2_2")
18   nodes_blade_surface = sample.get_nodes(base_name="Base_1_2")
19   elements_fluid = sample.get_elements(base_name="Base_2_2")
20   elements_blade_surface = sample.get_elements(base_name="Base_1_2")
21   nodal_tag_fluid = sample.get_nodal_tags(base_name="Base_2_2")
```

842  The meshes for the fluid domain and blade surface domain can also be converted to Muscat meshes:

```
1    from Muscat.Bridges import CGNSBridge
2    CGNS_tree = sample.get_mesh()
3    mesh_fluid = CGNSBridge.CGNSToMesh(CGNS_tree, baseNames=["Base_2_2"])
4    mesh_blade = CGNSBridge.CGNSToMesh(CGNS_tree, baseNames=["Base_1_2"])
```

843  ### B.3  2D_ElPlDynamics

844  2D_ElPlDynamics contains additional complexity: time-dependent data and a field located at the
845  center of the elements. When retrieving data, the default location of the fields is at the vertices. For
846  other type of fields, location mush be specified. Furthermore, in 2D_ElPlDynamics, the mesh is
847  different from one sample to another, but stays constant through the time sequence within a sample.
848  Hence, to prevent useless duplication of data, we link the geometrical support of the second to last
849  time step data to the mesh of the first time step. The corresponding commands are provided below:

```
1    from datasets import load_dataset
2    from plaid.containers.sample import Sample
3    import pickle
4
5    # Load the first sample of the train split
6    hf_dataset = load_dataset("PLAID-datasets/2D_ElastoPlastoDynamics",
     ↪ split="all_samples")
7    ids_train = hf_dataset.description["split"]["train"]
8    sample = Sample.model_validate(pickle.loads(hf_dataset[ids_train[0]]["sample"]))
9
10   # Examples of data retrievals
11   time_steps = sample.get_all_mesh_times()
12
13   for time in time_steps:
14       for fn in ["U_x","U_y"]:
15           field = sample.get_field(fn, time = time)
16       field = sample.get_field("EROSION_STATUS", location="CellCenter", time = time)
17
18   CGNS_tree_t0 = sample.get_mesh(time = 0.)
19   CGNS_tree_t1 = sample.get_mesh(time = 0.01, apply_links = True, in_memory = True)
```

850  ## C   Benchmarking online applications

851  Anyone wishing to participate in our benchmarks, hosted at huggingface.co/PLAIDcompetitions,
852  should create a Hugging Face account. However, no account is required to browse the website or view

the leaderboards. To participate, users simply need to train their model independently and submit predictions on the testing set. We do not require participants to upload their models. Two separate leaderboards are maintained, each based on a hidden subset of the test set, in order to discourage tentatives to overfit on the testing set.

We illustrate the benchmarking application using the `VKI-LS59` dataset as an example.

Figure 12 shows the benchmark homepage. A navigation menu is available on the left-hand side, allowing users to browse the site and log in. This page also provides examples of the dataset output fields and includes a visualization tool, where users can select a training sample ID and an output field to display.

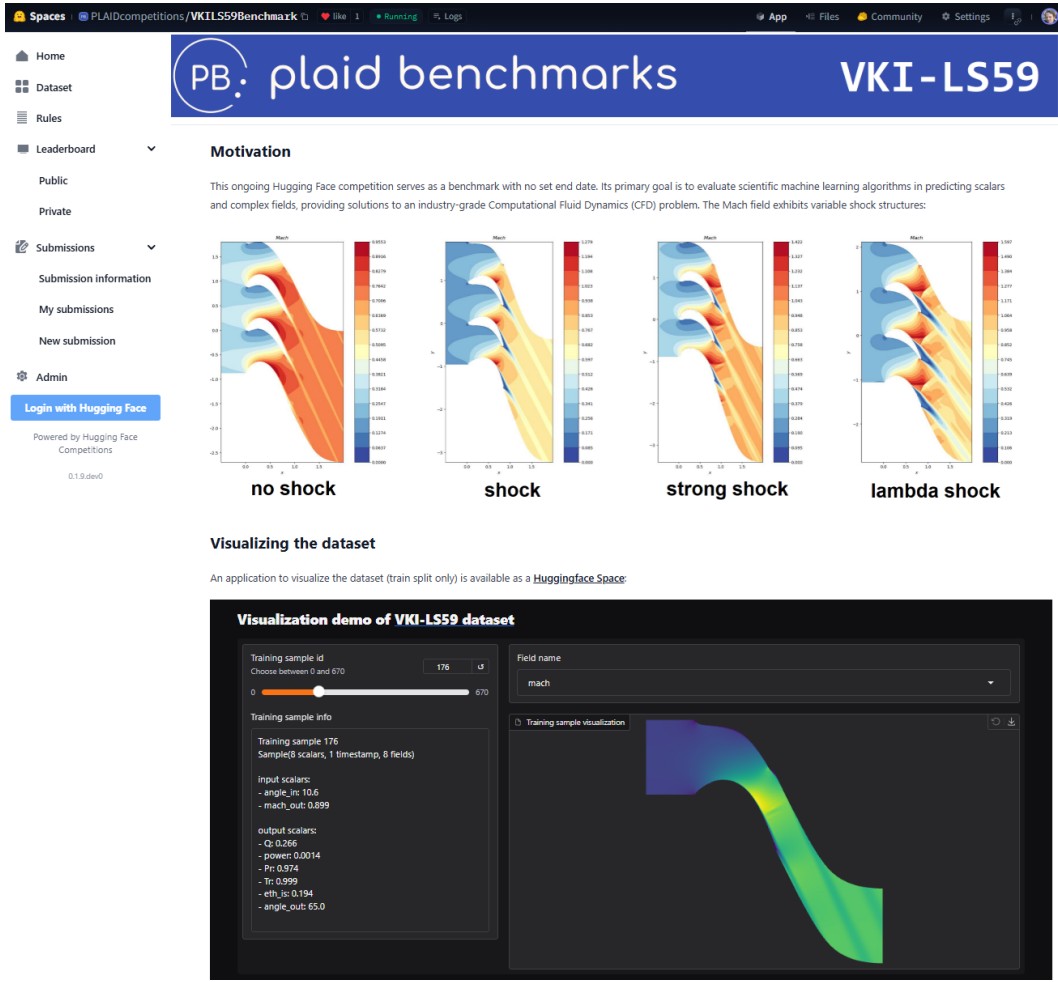

**Figure 12:** "Home" page of the benchmarking application on the `VKI-LS59` dataset.

Figure 13 provides detailed instructions on how to retrieve the dataset, including a description of the inputs and outputs used in the benchmark. Example commands are also provided to retrieve the samples and the required associated data.

The set of rules applying to the benchmark is presented in Figure 14.

Figure 15 provides detailed instructions on how to generate and submit the prediction file. The scoring function used for evaluation is also described.

Figure 16 illustrates the user's submissions page and the submission interface.

Figure 17 shows the public leaderboard as it appeared at the time of submission of this work.

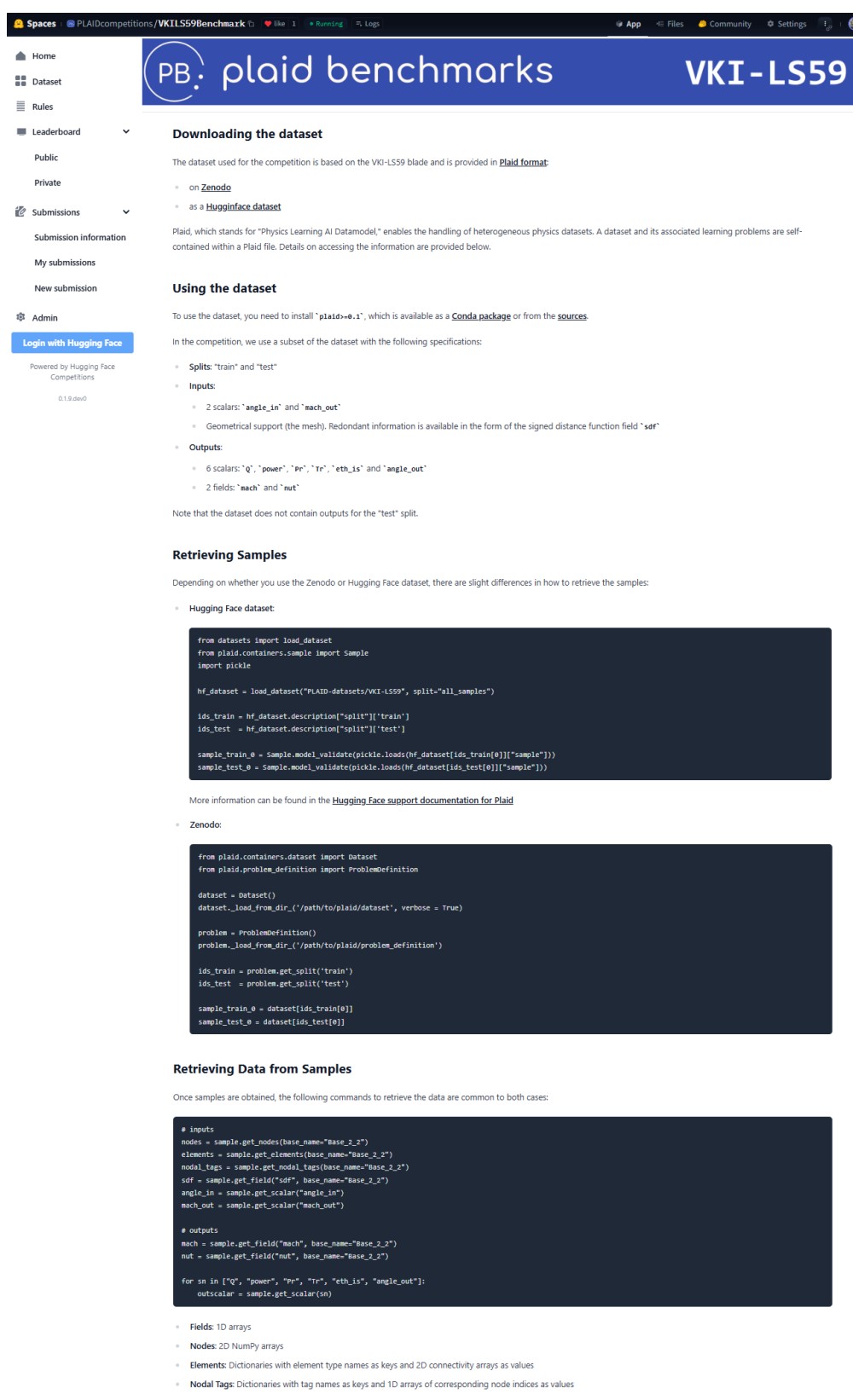

**Figure 13:** "Dataset" page of the benchmarking application on the VKI-LS59 dataset.

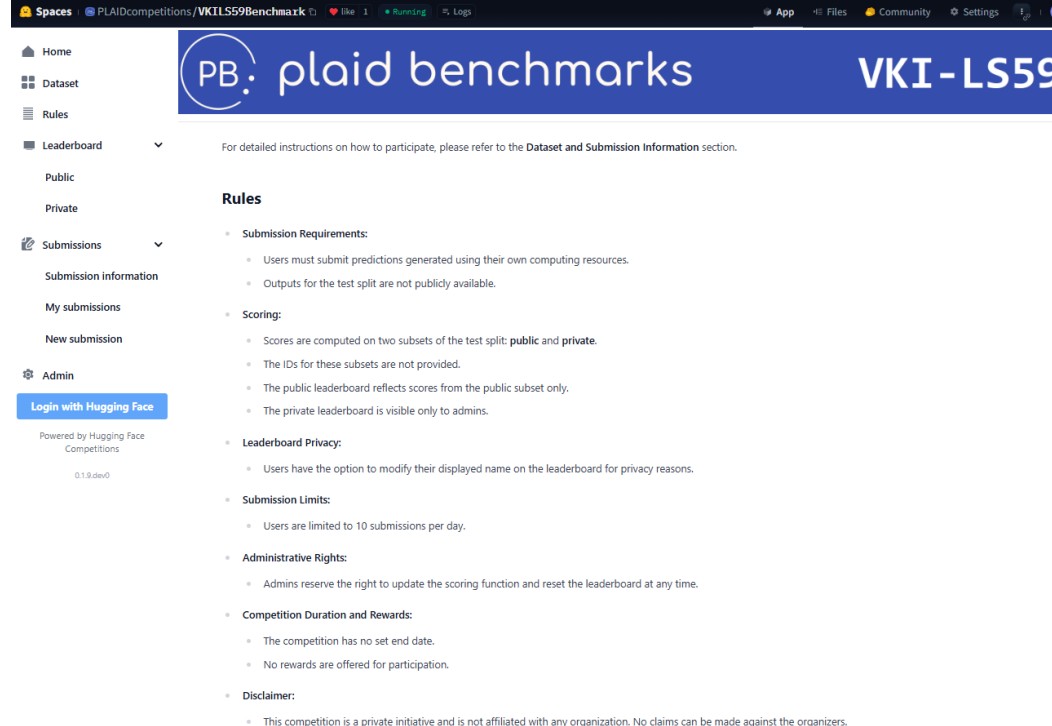

**Figure 14:** "Rules" page of the benchmarking application on the `VKI-LS59` dataset.

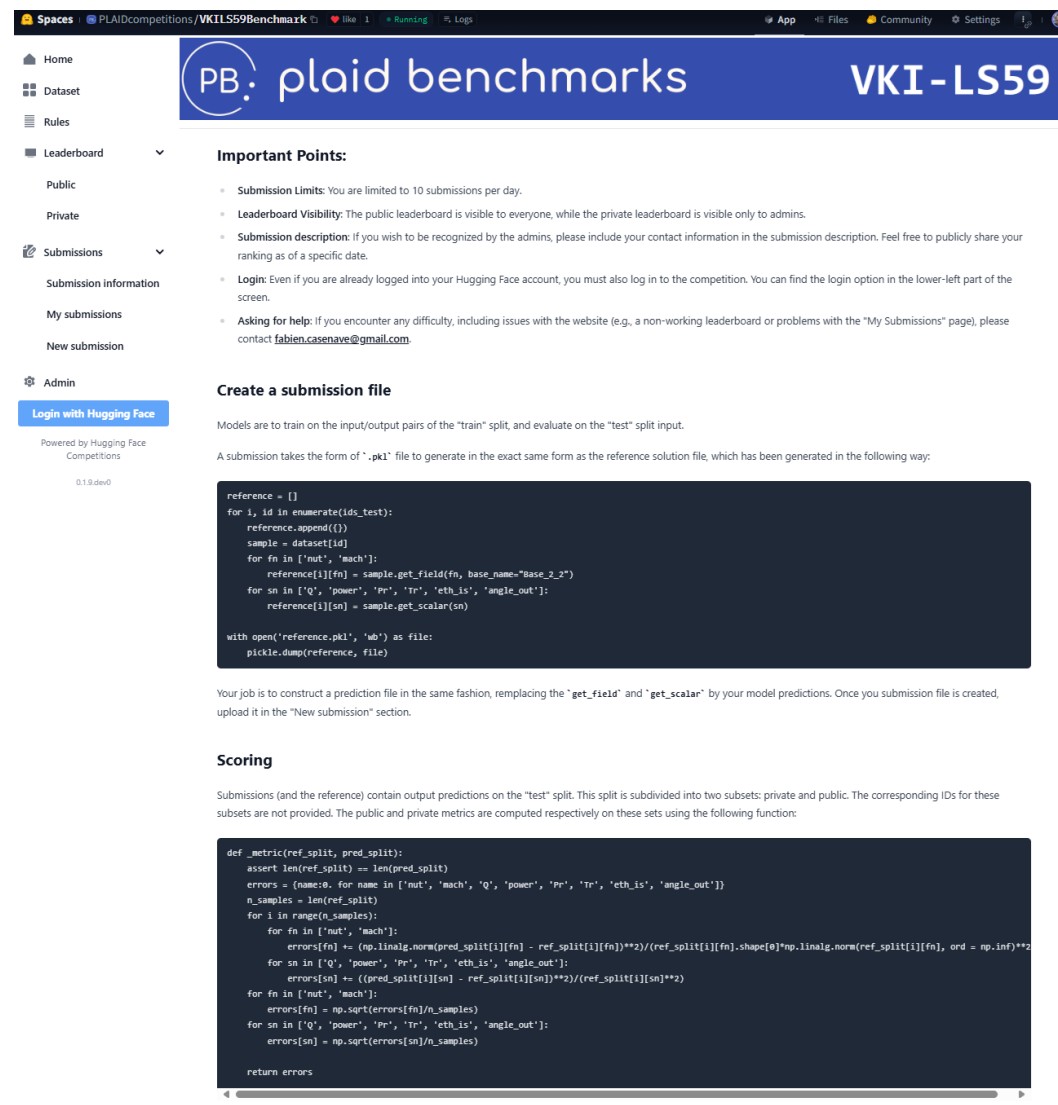

**Figure 15:** "Submission information" page of the benchmarking online application on the VKI-LS59 dataset.

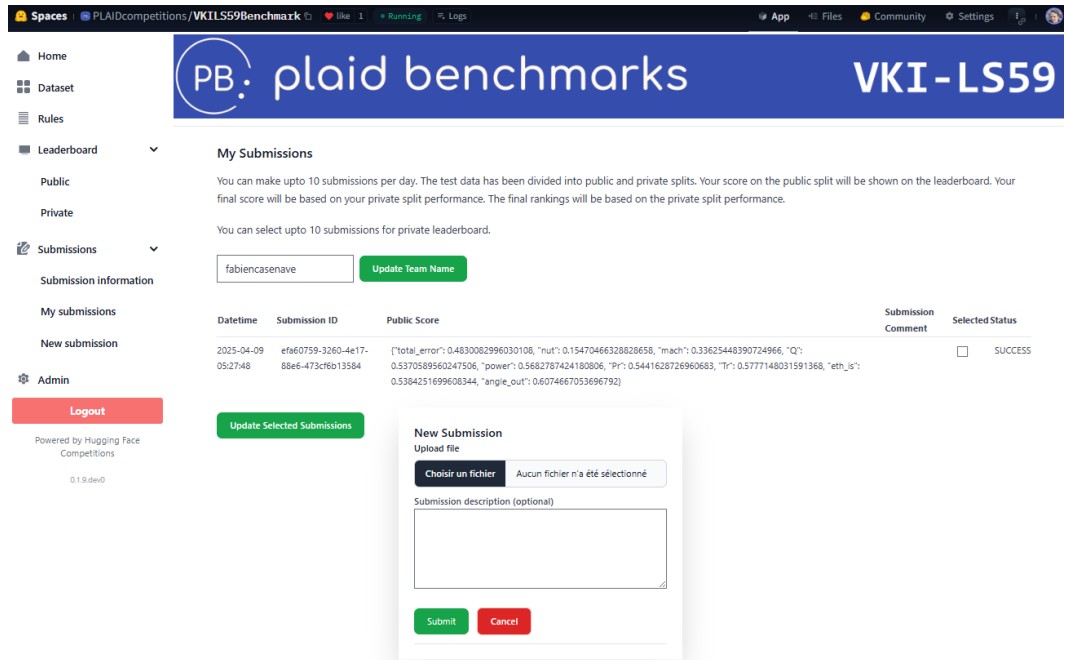

**Figure 16:** "My submissions" page of the benchmarking application on the VKI-LS59 dataset.

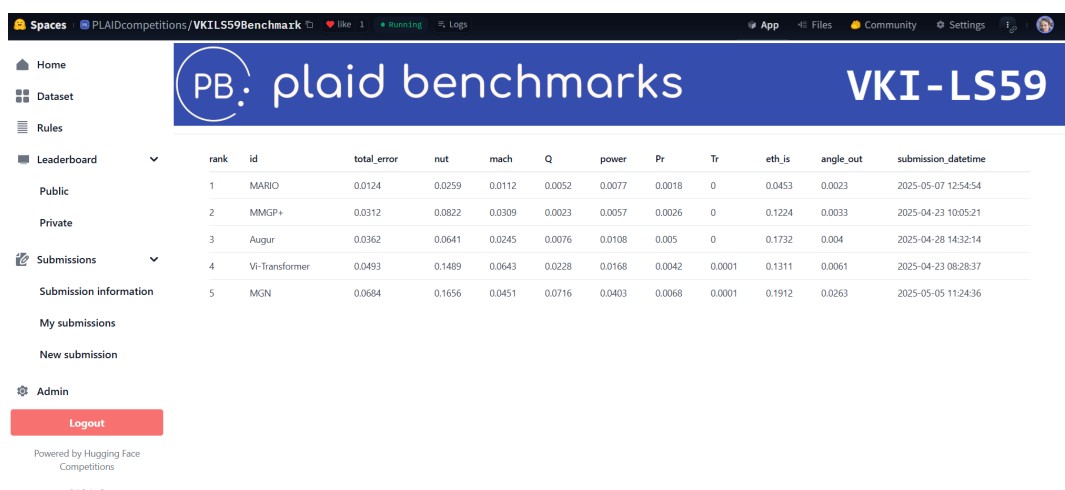

**Figure 17:** "Public leaderboard" page of the benchmarking application on the VKI-LS59 dataset.

