# OpenReview forum: "Physics-Learning AI Datamodel (PLAID) datasets: a collection of physics simulations for machine learning"
_NeurIPS.cc/2025/Datasets_and_Benchmarks_Track — Submitted to NeurIPS 2025 Datasets and Benchmarks Track_

### Official Review · Reviewer_Rgzi · 2025-06-24

**Rating:** 5
**Confidence:** 4

**Summary:**

In this paper, the authors present PLAID (Physics-Learning AI Datamodel), a flexible framework and library designed for representing and sharing physics simulation data. PLAID addresses a key challenge in the field: existing datasets are often limited in scope and rely on fragmented tools. To tackle this, the authors offer six datasets spanning structural mechanics and fluid dynamics, all featuring unstructured meshes—a format that is currently underrepresented and inadequately supported. Additionally, PLAID includes benchmarks and tools hosted on Hugging Face to encourage community involvement and facilitate standardized evaluation.

**Additional Feedback:**

* Line 102: The link labeled "online" is no longer working—please update it.

* Table 1: Define the variable “Nb” for clarity.

* Clarify the difference between the Zenodo and Hugging Face versions of the dataset. Why is there such a significant difference in dataset size as shown in Table 1?

* Table 12: The hardware and training time details are missing. These are essential for evaluating the experimental setup.

* Section 6: The discussion is minimal and should be expanded. Consider moving Table 3 to the appendix to create space for a more thorough discussion.

**Dataset Code Accessibility:**

Partly

**Dataset Code Comments:**

The dataset access and preprocessing tools are well-prepared. However, for full reproducibility, it would be beneficial to include example training scripts, particularly those used to produce the paper’s results.

**Ethical Comments:**

n.a

**Ethical Considerations:**

No, there are no or only very minor ethics concerns

**Final Justification:**

In this paper, the authors introduce PLAID (Physics-Learning AI Datamodel), a flexible framework and library designed to facilitate the representation and sharing of physics simulation data. PLAID addresses a significant challenge in the field: existing datasets are often limited in scope and depend on fragmented, inconsistent tools.

While I initially had concerns regarding the reproducibility of the benchmarks and the completeness of the dataset, the authors have satisfactorily addressed these issues during the rebuttal process.

I now consider this paper to be a strong candidate for acceptance in the NeurIPS Datasets and Benchmarks track.

**Limitations Weaknesses:**

Major:

* W1: Although this is simply due to my insufficient understanding of the code, the GitHub repository does not include training scripts or model implementations, making it impossible to reproduce the results reported in Table 4.
* W2: Table 4 contains several blank entries, but the paper lacks sufficient explanation or justification for these omissions.

Minor:
* W3: The dataset primarily consists of 2D simulations; no 3D data is provided.
* W4: The number of data samples is relatively small (approximately 1,000).
* W5: It is unclear whether the PLAID dataset is entirely new or a curated collection of existing datasets. Based on Section 4, it appears to be the latter—this should be stated more explicitly.
* W6: The paper does not include the equations used to generate the data. While it refers to "a non-linear constitutive law," the actual equations should be provided, especially for applications like PINO loss [1] that require such details.
* W7: It would be helpful to include an assessment of computational cost—for example, the typical simulation time required to generate the test data, compared to the time needed to generate the training dataset, along with the total model training and inference times.

[1] Li, Zongyi, et al. "Physics-informed neural operator for learning partial differential equations." ACM/JMS Journal of Data Science 1.3 (2024): 1-27.

**Strengths Contributions:**

* S1: The code is publicly available on GitHub and conveniently distributed via PyPI and Conda.

* S2: The dataset is hosted on Hugging Face with concise and accessible documentation.

* S3: The data includes complex mesh structures and structural mechanics simulations, which are either unavailable or insufficiently represented in existing resources.

---

> ### Author Rebuttal · Authors · 2025-07-28
>
> We sincerely thank the reviewer for the detailed and constructive feedback.
>
> ## Introduction
>
> In this work, we introduce **PLAID**, a unified data model for physics simulation datasets, along with:
>
> -	an open-source implementation of the data model,
> -	six diverse datasets from solid and fluid mechanics, available on **Zenodo** and integrated into the **Hugging Face** ecosystem,
> -	corresponding interactive benchmarking applications hosted on Hugging Face Spaces.
>
> **Comparison with existing datasets.** Our goal is to support fair and reproducible surrogate modeling across heterogeneous simulation settings. The datasets are designed to reflect realistic industrial conditions, including complex geometries, variable topologies, and diverse physical regimes. The datasets were generated using simulation codes employed in industry (Z-set, OpenRadioss, elsA, and BROADCAST), along with constitutive laws for solids and turbulence models that are used in industrial settings. To position PLAID within the landscape of existing dataset collections, we include the comparison below:
> | Dataset collection | Stationary | Unstationary | 2D+3D     | Complex domains | Geometrical variations | Complex mesh settings and sample heterogeneity | Addressed physics                                  |
> |--------------------|-----------------|------------------------|-----------|----------------------------|------------|--------------|----------------------------------------------------|
> | `Mechanical-MNIST` | ✅             | ✅                    | ❌        | ❌                         | ❌        | ❌           | Solids  |
> | `PDEArena`         | ❌             | ✅                    | ✅         | ❌                         | ❌        | ❌          | Fluids  |
> | `BubbleML`         | ❌             | ✅                    | ✅        | ❌                         | ❌        | ❌           | Boiling |
> | `BLASTNet`         | ✅             | ✅                    | ✅        | ❌                         | ❌        | ❌           | Fluids  |
> | `PDEBench`         | ✅             | ✅                    | ✅        | ❌                         | ❌        | ❌           | Fluids  |
> | `The Well`         | ❌             | ✅                    | ✅        | ✅                         | ✅ (via density)  | ❌           | Fluids, astro, acoustics |
> | `PINNacle`         | ✅             | ✅                    | ✅        | ✅                         | ❌        | ❌           | Heat, fluids, waves |
> | `PLAID Datasets`   | ✅             | ✅                    | ✅        | ✅                         | ✅        | ✅           | Fluids and solids    |
>
> References for these datasets will be provided in the revision.
>
> > **_NOTE:_** “Mesh/sample heterogeneity” refers to possible presence of multiple meshes per sample, with differing dimensions, topologies, element types and variable topology. For example, the `VKI-LS59` dataset contains two geometrical supports of different dimensionality per sample, and the `2D_ElPlDynamics` contains samples of variable topology (see more details in Appendix B.2 and B.3).
>
>
> **Rationale for Dataset Selection.** We also include below the motivation behind each dataset, both in terms of machine learning challenges (e.g., heterogeneity, topology variations) and industrial relevance:
> | Dataset           | ML Challenge                                                | Industrial relevance                                                           |
> |-------------------|-------------------------------------------------------------|--------------------------------------------------------------------------------|
> | `Tensile2d`       | Unstructured mesh, Variable size fields, nonlinear laws | Metallic materials with complex ElastoViscoPlastic behavior  |
> | `2D_MultiScHypEl` | Unstructured mesh, Variable size fields, variable topology | Representative volume element for bi-materials, e.g. composite materials         |
> | `2D_ElPlDynamics` | Variable size, time-dependent, topology var., nonlin mat.   | Structural integrity of components under extreme conditions                      |
> | `Rotor37`         | 3D, shocks of variable position, nonlinear model           | Design of rotors in compressors of rotating machinery                            |
> | `2D_profile`      | Var. size fields, var. shock number and positions, nonlin. | Design of wings and propellers                                                   |
> | `VKI-LS59`        | variable shock classes, periodic setting, nonlinear model  | Design of rotors in turbines of rotating machinery                               |
>
> > **_NOTE:_**  All these datasets have variable geometrical supports, which is a variability of prime interest for industrial design, where the shape of the parts and components play a major role.
>
> **Benchmark coverage.** We are also actively working to complete the benchmark entries (Table 4), some of which are already available on Hugging Face:
> | Dataset           | MGN | MMGP | Vi-Transf. | Augur | DAFNO | MARIO |
> |-------------------|-----|------|------------|-------|-------|-------|
> | `Tensile2d`       | 🔵  |  🔵  |   🔵     |  🔵   |   🕑  |  ✅   |
> | `2D_MultiScHypEl` | 🔵  |  ❌  |   🔵     |  🔵   |   🕑  |  ✅   |
> | `2D_ElPlDynamics` | 🕑  |  ❌  |   🕑     |  🕑    |  🔵  |  🕑   |
> | `Rotor37`         | 🔵  |  🔵  |   🔵     |  🔵   |   🕑🕑  |  ✅   |
> | `2D_profile`      | 🔵  |  🔵  |   🔵     |  ✅   |   🕑  |  ✅   |
> | `VKI-LS59`        | 🔵  |  🔵  |   🔵     |  🔵   |   🕑🕑  |  🔵  |
>
> - 🔵: Present in initial submission
> - ✅: Added post-submission on Hugging Face
> - ❌: Not compatible with topology variation
> - 🕑/🕑🕑: Work in progress
>
> The training scripts for all presented models will be provided — except for Augur, which is a commercial tool.
>
> Finally, we plan to extend the dataset collection with additional time-dependent simulations, including:
>
> -	crack propagation in structural mechanics, and
> -	unsteady turbulent CFD simulations,
>
> along with corresponding interactive benchmarks hosted on Hugging Face. The PLAID format already supports these scenarios. While we cannot guarantee their inclusion for the camera-ready version, we are fully committed to this roadmap. We believe now is the right time to introduce the PLAID standard, library, initial datasets, benchmark results, and interactive benchmarking applications to the community.
>
> ## Rebuttal
>
> **W1.** The repository currently does not include the trained models. We will provide scripts to reproduce all results from Table 4 online, except for the Augur model, which is a commercial solution.
>
> **W2.** We are actively working to complete the benchmark entries in Table 4, with the status detailed above. We are currently preparing new datasets, including crack propagation in structural mechanics and unsteady turbulent flow simulations.
>
> **W3.** The `Rotor37` dataset is indeed 3D. The Hugging Face visualization widget on the dataset page allows interactive 3D rotation. We restricted the field outputs to the blade surface for memory efficiency, but the skin is embedded in 3D space, hence coordinate fields remain 3-component vectors.
>
> **W4.** We agree that dataset size is important. While tabular/image/text datasets often feature millions of samples, scientific datasets are more constrained. While some existing datasets are very large, our datasets appear in line with many recent contributions. For stationary datasets, `AirfRANS` and `DrivaerML` are contributions of one dataset each, with 1,000 samples and 500 samples respectively. For time-dependent datasets, the Well series includes datasets from 9 to ~2,000 trajectories, with two cases having 8k and 10k. Our time-dependent dataset contains 1,018 trajectories, aligning well with community standards. On the contrary, `EAGLE` has more than 1 million samples but is a single dataset of ~3k nodes meshes, `MechanicalMNIST` is a collection with configurations of 70,000 samples, but on small structured meshes (28*28), `PDEBench` has datasets with 100, 1k or 10k samples, all the 10k-sample datasets being either 1D or 2D stationary.
>
> **W5.** Thank you for pointing this out. `Tensile2D`, `2D_MultiScHypEl`, and `Rotor37` were introduced in prior work. In contrast, `2D_ElPlDynamics`, `2D_profile`, and `VKI-LS59` are new contributions in this paper, as well as all the Hugging Face benchmarking initiatives. We will specify this in the revision.
>
> **W6.** We agree that models such as PINO require access to the governing physical equations in full form to compute residuals. However, providing high-fidelity simulation codes or full model specifications (e.g., constitutive laws for solids or turbulence models for fluids) would effectively allow anyone to reconstruct the physics-based numerical solvers and achieve near-perfect accuracy on the test sets. This would undermine the purpose of the interactive Hugging Face benchmarks, which are designed to evaluate learned surrogates in a fair and reproducible manner. To preserve the integrity of these benchmarks, we have intentionally withheld test set outputs and are cautious about releasing simulator-level details. Nonetheless, we will include the high-level governing equations in the revision to ensure clarity and completeness. We hope this compromise is acceptable and remain open to further discussion if the reviewer still finds this aspect critical for acceptance.
>
> **W7.** We will include dataset generation times, training times and inference times.
>
> **Additional Feedback.** We will fix the broken link, clarify variable definitions (e.g., `Nb` in Table 1), and complete Table 12 with hardware and training time. The discrepancies between Zenodo and Hugging Face are due to Hugging Face using an optimized dataset interface with Parquet format to profit from advanced hosting services (like partial download, streaming), while Zenodo hosts static tarballs. We will expand the discussion in Section 6 and update the benchmark section to include recently obtained results.

---

> > ### Comment · Reviewer_Rgzi · 2025-08-01
> > **reply**
> >
> > Thank you for your effort in addressing my concerns. My current evaluation is primarily based on the issues raised in points W1 and W2. Based on the author's response, I have the impression that the work is still in progress. Therefore, I will maintain my current score, but I strongly encourage the authors to focus on completing the dataset and code.
> >
> > Please note that I believe this work is both important and valuable. However, my evaluation reflects the current state of the submission, not its potential in a "far" future, more complete version.

---

> > > ### Author Response · Authors · 2025-08-05
> > > **Benchmark Finalized and Code Now Publicly Available**
> > >
> > > We thank the reviewer for their encouraging feedback and for responding early in the discussion period. This gave us both the opportunity and the motivation to complete the remaining work in time to provide a full response before the end of the discussion phase. As a result, we were able to finalize the benchmark table and release the code online. To improve reproducibility and alignment with common practice in the Neural Operator literature, we replaced DAFNO with the more widely cited FNO model, using the high-quality implementation available in NVIDIA’s PhysicsNemo library.
> > >
> > > We believe this additional work fully addresses points W1 and W2 raised in the initial review. The updated table is shown below (displaying only the `total_error`), and each entry corresponds to a submission that can be consulted on the Hugging Face interactive benchmarks. The code to reproduce these results (excluding **Augur**, which relies on a commercial solution and cannot be open-sourced) is available in the `benchmarks` folder of the PLAID repository.
> > >
> > > ### Benchmark results:
> > > | Dataset           | MGN | MMGP | Vi-Transf. | Augur | FNO | MARIO |
> > > |-------------------|-----|------|------------|-------|-------|-------|
> > > | `Tensile2d`       | 0.0673  |  **0.0026**  |   0.0116     |  0.0154   |  0.0123  |  *0.0038*  |
> > > | `2D_MultiScHypEl` | 0.0437  |  ❌  |   0.0325     |  **0.0232**   |   *0.0302*  |  0.0573  |
> > > | `2D_ElPlDynamics` | 0.1202  |  ❌  |   *0.0227*     |  0.0346    |  **0.0215**  |  0.0319  |
> > > | `Rotor37`         | 0.0074  |  **0.0014**  |   0.0029     |  0.0033   |   0.0313  |  *0.0017*  |
> > > | `2D_profile`      | 0.0593  |  0.0365  |   *0.0312*     |  0.0425   |  0.0972  |  **0.0307**  |
> > > | `VKI-LS59`        | 0.0684  |  0.0312  |   *0.0193*     |  0.0267    |   0.0215  |  **0.0124**  |
> > >
> > > ❌: Not compatible with topology variation
> > >
> > > ### Additional notes:
> > > - **MMGP** does not support variable mesh topologies, which limits its applicability to certain datasets and often necessitates custom preprocessing for new cases. However, when morphing is either unnecessary or inexpensive, it offers a highly efficient solution, combining fast training with good accuracy (e.g., `Tensile2d` and `Rotor37`).
> > > - **MARIO** is computationally expensive to train but achieves consistently a very strong performance across most datasets. Its result on `2D_MultiScHypEl` in slightly worse than other tested methods, which may reflect the challenge of capturing complex shape variability in these cases.
> > > - **Vi-Transformer** and **Augur** perform well across all datasets, showing strong versatility and generalization capabilities.
> > > - **FNO** suffers on datasets featuring unstructured meshes with pronounced anisotropies, due to the loss of accuracy introduced by projections to and from regular grids (e.g., `Rotor37` and `2D_profile`). Additionally, the use of a 3D regular grid on `Rotor37` results in substantial computational overhead.

---

> > > > ### Comment · Reviewer_Rgzi · 2025-08-06
> > > > **reply**
> > > >
> > > > Thank you to the authors for their hard work in addressing my concerns. Although I am not entirely confident whether the updated results and GitHub code fully comply with this year’s complex NeurIPS rules, the revisions have clearly resolved the key issues I raised. Therefore, I now consider this paper a strong candidate for acceptance and have increased my score accordingly.
> > > >
> > > > The following is my final feedback (not a rebuttal request):
> > > > To further improve the reproducibility of the work, I recommend providing pretrained model weights. This would be particularly helpful for newcomers to the community.

---

### Official Review · Reviewer_uYkB · 2025-06-30

**Rating:** 4
**Confidence:** 2

**Summary:**

This paper introduces a novel framework, PLAID, aimed at standardizing and facilitating the sharing of physics-based simulation datasets for machine learning applications. It addresses a critical bottleneck in the widespread adoption of ML-based surrogate models, which has been hindered by the scarcity of large-scale, diverse, and standardized datasets tailored for physics simulations. PLAID establishes a unified data standard based on CGNS, accompanied by a library for efficient data creation, reading, and manipulation across various physical use cases. The paper releases six carefully curated datasets covering structural mechanics and computational fluid dynamics, along with baseline benchmarks using representative learning methods, with tools available on Hugging Face for community participation and ongoing evaluation.

**Dataset Code Accessibility:**

Yes

**Dataset Code Comments:**

I have confirmed that the underlying code and data access for the PLAID dataset are generally functional. However, the meta-report indicates that the following Responsible AI (RAI) fields are missing.  I recommend that the authors address these omissions to enhance the dataset's transparency and adherence to responsible AI practices.

**Ethical Considerations:**

No, there are no or only very minor ethics concerns

**Final Justification:**

The authors have addressed my concerns thoroughly and sincerely. Upon reviewing their rebuttal, I am satisfied that the initial weaknesses. Given this, I believe the strengths of the paper outweigh its previous weaknesses, and I am comfortable maintaining my original score.

That said, I am not an expert in the physics-learning AI community. My research has primarily focused on specific areas such as PINNs and PINOs, so I assign a confidence level of 2. This is the only reason I am not increasing my rating further

**Limitations Weaknesses:**

- **Insufficient Comparison with Related Work**: The paper reviews existing datasets and benchmarks, noting their limitations such as focus on specific domains or fragmented formats. However, it could benefit from a more direct and explicit comparative analysis illustrating how PLAID specifically overcomes these limitations and offers superior advantages. A dedicated section or table highlighting PLAID's distinct features and improvements compared to prominent benchmarks like PDEBench , PDEArena , BubbleML , or The Well  would clarify its unique value proposition and help readers quickly grasp why PLAID is a more comprehensive and advantageous solution.

- **Ambiguous Rationale for Dataset Selection**: The paper presents six diverse datasets but lacks a clear and explicit justification for why these specific ones were chosen. While it mentions that they "showcase rich variability in physics and numerical complexity", the underlying criteria for their selection as a core benchmark set are not fully articulated. For a benchmark initiative, it is crucial to explain how each dataset is strategically designed to efficiently test specific algorithms or hypotheses. For instance, clarifying which dataset is particularly challenging for generalization, which highlights the importance of certain architectural biases, or which best represents a particular class of physical phenomena or numerical challenges would significantly enhance the dataset's utility as a "key dataset" for evaluating novel machine learning approaches.

- **Real-World Application Scenarios of Datasets**: The paper states that the six provided datasets "showcase rich variability in physics and numerical complexity". It would be beneficial to elaborate with concrete examples on how each dataset can be mapped to real-world industrial problems or specific research questions.

**Strengths Contributions:**

While my research focuses on some specific physics-informed AI (PINN, PINO, and so on), I may not be an expert on the entire breadth of physics-learning AI or intimately familiar with all related literature. However, I agree with the authors' observation that securing large-scale, diverse, and standardized datasets is indeed a crucial topic that needs to be addressed. I believe the PLAID initiative represents a significant step forward in establishing a much-needed common ground for data representation and sharing.

The main strengths and contributions of this work are:

- **Flexible and Extensible Data Model**: PLAID's data model is robust and versatile, designed to support a wide array of complex use cases, including time-dependent problems, remeshing, mixed-element unstructured meshes, node/element tagging, and varying spatial dimensions and topologies. This flexibility is crucial for accommodating the inherent heterogeneity of physics simulations and supporting future developments in foundation models for physics.

- **Open and Community-Oriented Infrastructure**: The provision of benchmarking tools on Hugging Face, enabling direct community participation and continuous evaluation efforts, is highly commendable. This fosters transparency, reproducibility, and collaborative advancement, which are pillars of robust scientific progress.

- **Comprehensive Data Collection with Practical Relevance**: The release of six datasets, including both structural mechanics and computational fluid dynamics, looks important. These datasets seem to exhibit rich variability in physics and numerical complexity, reflecting real-world industrial relevance.

---

> ### Author Rebuttal · Authors · 2025-07-28
>
> We sincerely thank the reviewer for the detailed and constructive feedback.
>
> ## Introduction
>
> In this work, we introduce **PLAID**, a unified data model for physics simulation datasets, along with:
>
> -	an open-source implementation of the data model,
> -	six diverse datasets from solid and fluid mechanics, available on **Zenodo** and integrated into the **Hugging Face** ecosystem,
> -	corresponding interactive benchmarking applications hosted on Hugging Face Spaces.
>
> **Comparison with existing datasets.** Our goal is to support fair and reproducible surrogate modeling across heterogeneous simulation settings. The datasets are designed to reflect realistic industrial conditions, including complex geometries, variable topologies, and diverse physical regimes. The datasets were generated using simulation codes employed in industry (Z-set, OpenRadioss, elsA, and BROADCAST), along with constitutive laws for solids and turbulence models that are used in industrial settings. To position PLAID within the landscape of existing dataset collections, we include the comparison below:
> | Dataset collection | Stationary | Unstationary | 2D+3D     | Complex domains | Geometrical variations | Complex mesh settings and sample heterogeneity | Addressed physics                                  |
> |--------------------|-----------------|------------------------|-----------|----------------------------|------------|--------------|----------------------------------------------------|
> | `Mechanical-MNIST` | ✅             | ✅                    | ❌        | ❌                         | ❌        | ❌           | Solids  |
> | `PDEArena`         | ❌             | ✅                    | ✅         | ❌                         | ❌        | ❌          | Fluids  |
> | `BubbleML`         | ❌             | ✅                    | ✅        | ❌                         | ❌        | ❌           | Boiling |
> | `BLASTNet`         | ✅             | ✅                    | ✅        | ❌                         | ❌        | ❌           | Fluids  |
> | `PDEBench`         | ✅             | ✅                    | ✅        | ❌                         | ❌        | ❌           | Fluids  |
> | `The Well`         | ❌             | ✅                    | ✅        | ✅                         | ✅ (via density)  | ❌           | Fluids, astro, acoustics |
> | `PINNacle`         | ✅             | ✅                    | ✅        | ✅                         | ❌        | ❌           | Heat, fluids, waves |
> | `PLAID Datasets`   | ✅             | ✅                    | ✅        | ✅                         | ✅        | ✅           | Fluids and solids    |
>
> References for these datasets will be provided in the revision.
>
> > **_NOTE:_** “Mesh/sample heterogeneity” refers to possible presence of multiple meshes per sample, with differing dimensions, topologies, element types and variable topology. For example, the `VKI-LS59` dataset contains two geometrical supports of different dimensionality per sample, and the `2D_ElPlDynamics` contains samples of variable topology (see more details in Appendix B.2 and B.3).
>
>
> **Rationale for Dataset Selection.** We also include below the motivation behind each dataset, both in terms of machine learning challenges (e.g., heterogeneity, topology variations) and industrial relevance:
> | Dataset           | ML Challenge                                                | Industrial relevance                                                           |
> |-------------------|-------------------------------------------------------------|--------------------------------------------------------------------------------|
> | `Tensile2d`       | Unstructured mesh, Variable size fields, nonlinear laws | Metallic materials with complex ElastoViscoPlastic behavior  |
> | `2D_MultiScHypEl` | Unstructured mesh, Variable size fields, variable topology | Representative volume element for bi-materials, e.g. composite materials         |
> | `2D_ElPlDynamics` | Variable size, time-dependent, topology var., nonlin mat.   | Structural integrity of components under extreme conditions                      |
> | `Rotor37`         | 3D, shocks of variable position, nonlinear model           | Design of rotors in compressors of rotating machinery                            |
> | `2D_profile`      | Var. size fields, var. shock number and positions, nonlin. | Design of wings and propellers                                                   |
> | `VKI-LS59`        | variable shock classes, periodic setting, nonlinear model  | Design of rotors in turbines of rotating machinery                               |
>
> > **_NOTE:_**  All these datasets have variable geometrical supports, which is a variability of prime interest for industrial design, where the shape of the parts and components play a major role.
>
> **Benchmark coverage.** We are also actively working to complete the benchmark entries (Table 4), some of which are already available on Hugging Face:
> | Dataset           | MGN | MMGP | Vi-Transf. | Augur | DAFNO | MARIO |
> |-------------------|-----|------|------------|-------|-------|-------|
> | `Tensile2d`       | 🔵  |  🔵  |   🔵     |  🔵   |   🕑  |  ✅   |
> | `2D_MultiScHypEl` | 🔵  |  ❌  |   🔵     |  🔵   |   🕑  |  ✅   |
> | `2D_ElPlDynamics` | 🕑  |  ❌  |   🕑     |  🕑    |  🔵  |  🕑   |
> | `Rotor37`         | 🔵  |  🔵  |   🔵     |  🔵   |   🕑🕑  |  ✅   |
> | `2D_profile`      | 🔵  |  🔵  |   🔵     |  ✅   |   🕑  |  ✅   |
> | `VKI-LS59`        | 🔵  |  🔵  |   🔵     |  🔵   |   🕑🕑  |  🔵  |
>
> - 🔵: Present in initial submission
> - ✅: Added post-submission on Hugging Face
> - ❌: Not compatible with topology variation
> - 🕑/🕑🕑: Work in progress
>
> The training scripts for all presented models will be provided — except for Augur, which is a commercial tool.
>
> Finally, we plan to extend the dataset collection with additional time-dependent simulations, including:
>
> -	crack propagation in structural mechanics, and
> -	unsteady turbulent CFD simulations,
>
> along with corresponding interactive benchmarks hosted on Hugging Face. The PLAID format already supports these scenarios. While we cannot guarantee their inclusion for the camera-ready version, we are fully committed to this roadmap. We believe now is the right time to introduce the PLAID standard, library, initial datasets, benchmark results, and interactive benchmarking applications to the community.
>
> ## Rebuttal
>
> **Insufficient Comparison with Related Work.** We agree that a more explicit comparison would strengthen the paper. We will revise Section 2 to include the comparison with existing dataset collections.
>
> **Ambiguous Rationale for Dataset Selection and Real-World Application Scenarios of Datasets.** We acknowledge that the rationale for the dataset selection needs to be clarified. While Section 4 currently provides a brief description of each setting and its inputs/outputs, we agree that it does not clearly articulate the dual motivation from both a machine learning and application perspective. We will include the motivation behind each dataset, as presented above.
>
> **Responsible AI fields.** Thank you for this helpful observation. We followed the NeurIPS 2025 Data Hosting Guidelines, which describe how to retrieve automatically generated Croissant metadata files for datasets hosted on Hugging Face. We will investigate what additional configurations are required on Hugging Face to ensure that Responsible AI (RAI) metadata fields are properly populated and visible.

---

> > ### Comment · Reviewer_uYkB · 2025-08-05
> >
> > I appreciate the authors' detailed and sincere responses to the reviewers' feedback.  I have no further questions.

---

### Official Review · Reviewer_jGso · 2025-07-02

**Rating:** 4
**Confidence:** 3

**Summary:**

The paper introduce PLAID, an open-source benchmark suite for physics-learning surrogate models that contains six high-resolution datasets under a common datamodel, provides baseline implementations and training scripts, and hosts a public Hugging Face leaderboard to facilitate reproducible comparisons and community contributions.

**Dataset Code Accessibility:**

Yes

**Ethical Considerations:**

No, there are no or only very minor ethics concerns

**Final Justification:**

I keep the current scores since the mainly rebuttal is based on future work.

**Limitations Weaknesses:**

- The benchmark currently includes only six datasets, which is limited to structural mechanics and computational fluid-dynamics with no multi-physics examples. Although the authors state that they will add larger, more diverse, industry-relevant datasets, it remains unclear how feasible this is and how quickly updates will arrive.

- Many model–dataset results in Table 4 are untested; for example, 2D_ElPlDynamics is evaluated with only a single model. The authors should provide runnable baseline implementations for every model so that each table entry has a value, even if performance is modest. They could then encourage community pull requests via the Hugging Face leaderboard, establish a clear submission deadline, and provide a complete benchmark that future work can cite.

**Strengths Contributions:**

- It combines six high-quality physics datasets into one unified schema, so users can compare models directly without extra data-cleaning work.

- The simulations keep fine spatial and temporal resolution, and the tasks resemble real engineering difficulty.

- A public Hugging Face leaderboard records every submission. This gives transparent ranking and motivating community contribution.

---

> ### Author Rebuttal · Authors · 2025-07-28
>
> We sincerely thank the reviewer for the detailed and constructive feedback.
>
> ## Introduction
>
> In this work, we introduce **PLAID**, a unified data model for physics simulation datasets, along with:
>
> -	an open-source implementation of the data model,
> -	six diverse datasets from solid and fluid mechanics, available on **Zenodo** and integrated into the **Hugging Face** ecosystem,
> -	corresponding interactive benchmarking applications hosted on Hugging Face Spaces.
>
> **Comparison with existing datasets.** Our goal is to support fair and reproducible surrogate modeling across heterogeneous simulation settings. The datasets are designed to reflect realistic industrial conditions, including complex geometries, variable topologies, and diverse physical regimes. The datasets were generated using simulation codes employed in industry (Z-set, OpenRadioss, elsA, and BROADCAST), along with constitutive laws for solids and turbulence models that are used in industrial settings. To position PLAID within the landscape of existing dataset collections, we include the comparison below:
> | Dataset collection | Stationary | Unstationary | 2D+3D     | Complex domains | Geometrical variations | Complex mesh settings and sample heterogeneity | Addressed physics                                  |
> |--------------------|-----------------|------------------------|-----------|----------------------------|------------|--------------|----------------------------------------------------|
> | `Mechanical-MNIST` | ✅             | ✅                    | ❌        | ❌                         | ❌        | ❌           | Solids  |
> | `PDEArena`         | ❌             | ✅                    | ✅         | ❌                         | ❌        | ❌          | Fluids  |
> | `BubbleML`         | ❌             | ✅                    | ✅        | ❌                         | ❌        | ❌           | Boiling |
> | `BLASTNet`         | ✅             | ✅                    | ✅        | ❌                         | ❌        | ❌           | Fluids  |
> | `PDEBench`         | ✅             | ✅                    | ✅        | ❌                         | ❌        | ❌           | Fluids  |
> | `The Well`         | ❌             | ✅                    | ✅        | ✅                         | ✅ (via density)  | ❌           | Fluids, astro, acoustics |
> | `PINNacle`         | ✅             | ✅                    | ✅        | ✅                         | ❌        | ❌           | Heat, fluids, waves |
> | `PLAID Datasets`   | ✅             | ✅                    | ✅        | ✅                         | ✅        | ✅           | Fluids and solids    |
>
> References for these datasets will be provided in the revision.
>
> > **_NOTE:_** “Mesh/sample heterogeneity” refers to possible presence of multiple meshes per sample, with differing dimensions, topologies, element types and variable topology. For example, the `VKI-LS59` dataset contains two geometrical supports of different dimensionality per sample, and the `2D_ElPlDynamics` contains samples of variable topology (see more details in Appendix B.2 and B.3).
>
>
> **Rationale for Dataset Selection.** We also include below the motivation behind each dataset, both in terms of machine learning challenges (e.g., heterogeneity, topology variations) and industrial relevance:
> | Dataset           | ML Challenge                                                | Industrial relevance                                                           |
> |-------------------|-------------------------------------------------------------|--------------------------------------------------------------------------------|
> | `Tensile2d`       | Unstructured mesh, Variable size fields, nonlinear laws | Metallic materials with complex ElastoViscoPlastic behavior  |
> | `2D_MultiScHypEl` | Unstructured mesh, Variable size fields, variable topology | Representative volume element for bi-materials, e.g. composite materials         |
> | `2D_ElPlDynamics` | Variable size, time-dependent, topology var., nonlin mat.   | Structural integrity of components under extreme conditions                      |
> | `Rotor37`         | 3D, shocks of variable position, nonlinear model           | Design of rotors in compressors of rotating machinery                            |
> | `2D_profile`      | Var. size fields, var. shock number and positions, nonlin. | Design of wings and propellers                                                   |
> | `VKI-LS59`        | variable shock classes, periodic setting, nonlinear model  | Design of rotors in turbines of rotating machinery                               |
>
> > **_NOTE:_**  All these datasets have variable geometrical supports, which is a variability of prime interest for industrial design, where the shape of the parts and components play a major role.
>
> **Benchmark coverage.** We are also actively working to complete the benchmark entries (Table 4), some of which are already available on Hugging Face:
> | Dataset           | MGN | MMGP | Vi-Transf. | Augur | DAFNO | MARIO |
> |-------------------|-----|------|------------|-------|-------|-------|
> | `Tensile2d`       | 🔵  |  🔵  |   🔵     |  🔵   |   🕑  |  ✅   |
> | `2D_MultiScHypEl` | 🔵  |  ❌  |   🔵     |  🔵   |   🕑  |  ✅   |
> | `2D_ElPlDynamics` | 🕑  |  ❌  |   🕑     |  🕑    |  🔵  |  🕑   |
> | `Rotor37`         | 🔵  |  🔵  |   🔵     |  🔵   |   🕑🕑  |  ✅   |
> | `2D_profile`      | 🔵  |  🔵  |   🔵     |  ✅   |   🕑  |  ✅   |
> | `VKI-LS59`        | 🔵  |  🔵  |   🔵     |  🔵   |   🕑🕑  |  🔵  |
>
> - 🔵: Present in initial submission
> - ✅: Added post-submission on Hugging Face
> - ❌: Not compatible with topology variation
> - 🕑/🕑🕑: Work in progress
>
> The training scripts for all presented models will be provided — except for Augur, which is a commercial tool.
>
> Finally, we plan to extend the dataset collection with additional time-dependent simulations, including:
>
> -	crack propagation in structural mechanics, and
> -	unsteady turbulent CFD simulations,
>
> along with corresponding interactive benchmarks hosted on Hugging Face. The PLAID format already supports these scenarios. While we cannot guarantee their inclusion for the camera-ready version, we are fully committed to this roadmap. We believe now is the right time to introduce the PLAID standard, library, initial datasets, benchmark results, and interactive benchmarking applications to the community.
>
> ## Rebuttal
>
> **Scope and future datasets.** We fully agree that increasing physics diversity and including multiphysics problems is essential. As mentioned above, we are currently preparing new datasets, including crack propagation in structural mechanics and unsteady turbulent flow simulations.
>
> **Benchmark coverage (Table 4).** We are actively working to complete the benchmark entries in Table 4, with the status detailed above.
>
> **Leaderboard and community involvement.** The Hugging Face leaderboard will continue accepting community submissions after publication. We will clarify this in the paper and explicitly encourage contributions. We deliberately chose not to impose a deadline, as we aim for PLAID benchmarks to serve as lasting references rather than time-limited competitions. We believe that the current benchmark results offer a reproducible, citable reference for the first six datasets at submission time.

---

> > ### Comment · Reviewer_jGso · 2025-08-05
> >
> > Thank you for replying my concerns. I will keep my current scores.

---

### Official Review · Reviewer_gjYa · 2025-07-03

**Rating:** 4
**Confidence:** 3

**Summary:**

The authors introduce Physics-Learning AI Datamodel (PLAID), a framework for representing physics simulation datasets, with a focus towards benchmarking surrogate (especially ML) models. The authors demonstrate:
1. A CGNS based schema to store numerical simulation data in a standardized machine and human readable format, capable of handling heterogenous data.
2. Seven datasets covering structural mechanics and CFD simulations, available on Hugging Face and Zenodo.
3. Benchmark results for five ML models on these datasets.
Overall, this work is useful for practitioners working on scientific ML (specifically PDE solvers) for benchmarking new methods on unstructured geometries.

**Dataset Code Accessibility:**

Partly

**Dataset Code Comments:**

I was unable to install the library with the instructions provided on the github repo:

First, on with my python 3.13, it didn't meet requirements:
```
ERROR: Ignored the following versions that require a different python version: 0.1.2 Requires-Python <3.13,>=3.11; 0.1.3 Requires-Python <3.13,>=3.11; 0.1.4 Requires-Python <3.13,>=3.11; 0.1.5 Requires-Python <3.13,>=3.11; 0.1.6 Requires-Python <3.13,>=3.9
ERROR: Could not find a version that satisfies the requirement pyplaid
```

I created a new python 3.12 env. I got errors about hdf5:
```
 ../CGNS/meson.build:113:11: ERROR: Dependency "hdf5" not found, tried pkgconfig, pkgconfig and config-tool
```

While building it, I got an error:
`../CGNS/meson.build:113:11: ERROR: Dependency "hdf5" not found, tried pkgconfig, pkgconfig and config-tool` which seems to be unrelated to the Muscat note (pypi vs conda-forge availability).

The pip package should be self-contained, and any dependencies and related installation instructions should be documented.

Apart from this, the Zenodo and Huggingface links appear to be correct and hosting the data. The project page is nice and includes interactive widgets to explore the datasets: https://plaid-lib.github.io/project/

**Ethical Considerations:**

No, there are no or only very minor ethics concerns

**Final Justification:**

The rebuttal satisfactorily resolves my smaller concerns: missing citations, reference formatting, broken links, and installation hurdles by outlining concrete fixes and demonstrating that the library has better environment documentation. It also clarifies that additional evaluation metrics and a more detailed limitations section will be added. Nevertheless, the central shortcoming remains: the benchmark suite is still largely static, with the promised time-dependent datasets and richer task-specific metrics only projected for future releases, and none of the new material is yet visible in the uploaded manuscript. Because the core contribution, a unified CGNS-based data model plus six datasets, is already valuable and technically solid, I continue to lean toward acceptance, but the above gaps keep my recommendation at 4.

**Limitations Weaknesses:**

- While the authors claim in line 57 that some existing datasets are isolated to time dependence, the datasets proposed in this paper are mostly for static systems (only one is transient `2D_ElPlDnamics`). The inclusion of more time-dependent system would strengthen the overall benchmark suite.
- While the authors mention "The Well" and other PDE benchmarks, I feel some comparable benchmarks such as CoDBench and Mechanical MNIST would be relevant in the related works section.
- Evaluation metrics (section 5.2) could be further strengthened with some task-specific observable errors in addition to broad field and scalar errors.
- The limitations and roadmap section is quite brief. Expanding the limitations and including a conclusions paragraph would improve the readability of this paper.
- There were some issues in trying to run the code. Please check Data Availability comments.

Minor points:
- The gitlab link seems to be archived. Please include the correct github links in the revised version.
- Some references are formatted inconsistently. Examples are the references for individual datasets [69, 71, 74, 76 78, 79, 55 81 - 83]. Please format them consistently, and add a doi or another accessible link to the references (not just in the subsection titles).
- This might be pedantic, but I cannot find many instances of the word datamodel without spaces. Please use "data models" if possible.

**Strengths Contributions:**

- The paper identifies clear gaps in existing scientific ML datasets and proposes a solution for them. This is relevant for engineering applications as most of the existing datasets focus on simpler toy problems.
- The framework proposed can handle heterogeneous data such as multilevel meshes, vertices/element tags etc.
- The datasets provided seem to be high quality 2D structural mechanics and 3D Navier-Stokes simulations, bigger in scale and complexity than standard PDEs used for benchmarking ML based solvers.
- Easy accessibility for both the datasets and the leaderboard through Zenodo and Hugging Face. The `pyplaid` library is also documented in detail and would allow for easy loading of these datasets.

---

> ### Author Rebuttal · Authors · 2025-07-28
>
> We sincerely thank the reviewer for the detailed and constructive feedback.
>
> ## Introduction
>
> In this work, we introduce **PLAID**, a unified data model for physics simulation datasets, along with:
>
> -	an open-source implementation of the data model,
> -	six diverse datasets from solid and fluid mechanics, available on **Zenodo** and integrated into the **Hugging Face** ecosystem,
> -	corresponding interactive benchmarking applications hosted on Hugging Face Spaces.
>
> **Comparison with existing datasets.** Our goal is to support fair and reproducible surrogate modeling across heterogeneous simulation settings. The datasets are designed to reflect realistic industrial conditions, including complex geometries, variable topologies, and diverse physical regimes. The datasets were generated using simulation codes employed in industry (Z-set, OpenRadioss, elsA, and BROADCAST), along with constitutive laws for solids and turbulence models that are used in industrial settings. To position PLAID within the landscape of existing dataset collections, we include the comparison below:
> | Dataset collection | Stationary | Unstationary | 2D+3D     | Complex domains | Geometrical variations | Complex mesh settings and sample heterogeneity | Addressed physics                                  |
> |--------------------|-----------------|------------------------|-----------|----------------------------|------------|--------------|----------------------------------------------------|
> | `Mechanical-MNIST` | ✅             | ✅                    | ❌        | ❌                         | ❌        | ❌           | Solids  |
> | `PDEArena`         | ❌             | ✅                    | ✅         | ❌                         | ❌        | ❌          | Fluids  |
> | `BubbleML`         | ❌             | ✅                    | ✅        | ❌                         | ❌        | ❌           | Boiling |
> | `BLASTNet`         | ✅             | ✅                    | ✅        | ❌                         | ❌        | ❌           | Fluids  |
> | `PDEBench`         | ✅             | ✅                    | ✅        | ❌                         | ❌        | ❌           | Fluids  |
> | `The Well`         | ❌             | ✅                    | ✅        | ✅                         | ✅ (via density)  | ❌           | Fluids, astro, acoustics |
> | `PINNacle`         | ✅             | ✅                    | ✅        | ✅                         | ❌        | ❌           | Heat, fluids, waves |
> | `PLAID Datasets`   | ✅             | ✅                    | ✅        | ✅                         | ✅        | ✅           | Fluids and solids    |
>
> References for these datasets will be provided in the revision.
>
> > **_NOTE:_** “Mesh/sample heterogeneity” refers to possible presence of multiple meshes per sample, with differing dimensions, topologies, element types and variable topology. For example, the `VKI-LS59` dataset contains two geometrical supports of different dimensionality per sample, and the `2D_ElPlDynamics` contains samples of variable topology (see more details in Appendix B.2 and B.3).
>
>
> **Rationale for Dataset Selection.** We also include below the motivation behind each dataset, both in terms of machine learning challenges (e.g., heterogeneity, topology variations) and industrial relevance:
> | Dataset           | ML Challenge                                                | Industrial relevance                                                           |
> |-------------------|-------------------------------------------------------------|--------------------------------------------------------------------------------|
> | `Tensile2d`       | Unstructured mesh, Variable size fields, nonlinear laws | Metallic materials with complex ElastoViscoPlastic behavior  |
> | `2D_MultiScHypEl` | Unstructured mesh, Variable size fields, variable topology | Representative volume element for bi-materials, e.g. composite materials         |
> | `2D_ElPlDynamics` | Variable size, time-dependent, topology var., nonlin mat.   | Structural integrity of components under extreme conditions                      |
> | `Rotor37`         | 3D, shocks of variable position, nonlinear model           | Design of rotors in compressors of rotating machinery                            |
> | `2D_profile`      | Var. size fields, var. shock number and positions, nonlin. | Design of wings and propellers                                                   |
> | `VKI-LS59`        | variable shock classes, periodic setting, nonlinear model  | Design of rotors in turbines of rotating machinery                               |
>
> > **_NOTE:_**  All these datasets have variable geometrical supports, which is a variability of prime interest for industrial design, where the shape of the parts and components play a major role.
>
> **Benchmark coverage.** We are also actively working to complete the benchmark entries (Table 4), some of which are already available on Hugging Face:
> | Dataset           | MGN | MMGP | Vi-Transf. | Augur | DAFNO | MARIO |
> |-------------------|-----|------|------------|-------|-------|-------|
> | `Tensile2d`       | 🔵  |  🔵  |   🔵     |  🔵   |   🕑  |  ✅   |
> | `2D_MultiScHypEl` | 🔵  |  ❌  |   🔵     |  🔵   |   🕑  |  ✅   |
> | `2D_ElPlDynamics` | 🕑  |  ❌  |   🕑     |  🕑    |  🔵  |  🕑   |
> | `Rotor37`         | 🔵  |  🔵  |   🔵     |  🔵   |   🕑🕑  |  ✅   |
> | `2D_profile`      | 🔵  |  🔵  |   🔵     |  ✅   |   🕑  |  ✅   |
> | `VKI-LS59`        | 🔵  |  🔵  |   🔵     |  🔵   |   🕑🕑  |  🔵  |
>
> - 🔵: Present in initial submission
> - ✅: Added post-submission on Hugging Face
> - ❌: Not compatible with topology variation
> - 🕑/🕑🕑: Work in progress
>
> The training scripts for all presented models will be provided — except for Augur, which is a commercial tool.
>
> Finally, we plan to extend the dataset collection with additional time-dependent simulations, including:
>
> -	crack propagation in structural mechanics, and
> -	unsteady turbulent CFD simulations,
>
> along with corresponding interactive benchmarks hosted on Hugging Face. The PLAID format already supports these scenarios. While we cannot guarantee their inclusion for the camera-ready version, we are fully committed to this roadmap. We believe now is the right time to introduce the PLAID standard, library, initial datasets, benchmark results, and interactive benchmarking applications to the community.
>
> ## Rebuttal
>
> **Time-dependent dataset.** We agree with the reviewer’s comment. As mentioned above, we are currently preparing new datasets, including crack propagation in structural mechanics and unsteady turbulent flow simulations.
>
> **Related work.** We will revise Section 2 to include the comparison with existing dataset collections.
>
> **Evaluation metrics.** It is possible to update the score functions with additional evaluations of the solution. We mention that all individual field and scalar scores detailed in Table 4 and on the Hugging Face benchmarks already contain task-specific quantities, for instance the max Von Mises stress in structural mechanics, the compression ratio and efficiency of the `Rotor37` blade, or the efficiency (`eth_is`) of the `VKI-LS59` blade. It is in our roadmap for PLAID to include evaluation metrics directly in the library, like proper L2 and energy error norms and Wasserstein distances, for fields.
>
> **Limitations and conclusion.** We plan to move Table 4 to the annex and to only keep in the main part of the paper a reduced Table with the `total_error` for each (dataset, method) couple. With the saved place, we will enrich Section 6, by mentioning future dataset (like the crack growth and turbulent CFD ones), and detailing the roadmap on evaluation metrics and pipelines (the ongoing pull request #101 on the repo already proposes a pipeline mechanism standardized for PLAID and satisfying the scikit-learn API).
>
> **Installability issues.** We thank the reviewer for having tried the library and for pointing this out. Likely, tests were made while we were in the process of configuring the github, the actions mechanisms, and the deployments on conda-forge and PyPi. As the reviewer has noticed, Muscat is currently only distributed via conda, but the development dependencies (including Muscat) are required for running tests and examples. We believe that the pip package is currently working, since we have checked that the following is working:
>
> ```python
> conda create -n testenv python=3.12 muscat=2.4.1         #(or mamba create, for faster install)
> conda activate testenv
> pip install pyplaid[dev]
> cd /path/to/plaid
> git checkout 0.1.6
> pytest tests                                             #243 tests should pass
> cd examples/
> ./run_examples.sh                                        #should return no error
> ```
>
> We configured a github action in the repo called `Tests and Examples`: an environment is configured daily from the repo’s `environment.yml` and all the tests and examples are run, for python 3.9 to 3.13, on linux, macOS and windows. The coverage is computed from the tests’ execution. We will update the README and the documentation to clarify that the pip package should not be used for tests and examples, and that a conda environment created from `environment.yml` is the preferred way of running tests and examples.
>
> **Minor points.** We will update the links with the github ones. We will conscientiously check all the references, in particular [69, 71, 74, 76 78, 79, 81 - 83], to include DOIs/links and all required information. We will update all instances of “datamodel(s)” to “data model(s)”.

---

> > ### Comment · Reviewer_gjYa · 2025-08-03
> >
> > The rebuttal satisfactorily resolves my smaller concerns: missing related-work citations, reference formatting, broken links, and the installation hurdles by outlining concrete fixes and demonstrating that the library now installs cleanly in a conda environment. It also clarifies that additional evaluation metrics and a more detailed limitations section will be added. Nevertheless, the central shortcoming remains: the benchmark suite is still largely static, with the promised time-dependent datasets and richer task-specific metrics only projected for future releases, and none of the new material is yet visible in the uploaded manuscript. Because the core contribution, a unified CGNS-based data model plus six datasets, is already valuable and technically solid, I continue to lean toward acceptance, but the above gaps keep my recommendation at 4.

---

### Note · Authors · 2025-08-12

We thank the reviewer for their time and constructive feedback.

We introduce PLAID, a unified data model for physics simulation datasets, together with:

- an open-source implementation of the data model, hosted on GitHub, with 100% test coverage, available via both PyPI and conda-forge,
- six diverse datasets from solid and fluid mechanics, including one time-dependent dataset, published on Zenodo and integrated into the Hugging Face ecosystem,
- a benchmark comparing six SciML methods on these datasets, with fully reproducible code publicly released,
- interactive benchmarking applications hosted on Hugging Face Spaces, allowing anyone to submit predictions and appear in a live leaderboard.

The datasets are designed to reflect realistic industrial conditions, incorporating complex geometries, variable topologies, and diverse physical regimes. They were generated using industry-grade simulation codes (Z-set, OpenRadioss, elsA, and BROADCAST) together with constitutive laws for solids and turbulence models commonly used in practice. Stationary datasets are also industrially relevant—particularly in the aerospace sector—where most designs are developed using quasistatic or stationary simulations. Each dataset was selected to highlight specific machine learning challenges while retaining strong industrial applicability, as detailed in the rebuttals.

To the best of our knowledge, the novelty of our work lies in:

- a CGNS-based data model enabling rich physical and geometrical descriptions (e.g., remeshing, element and node tags, multiple meshes per sample, anisotropies),
- datasets featuring complex geometrical variability and heterogeneous samples, now with an explicit comparison table against existing datasets (ready for inclusion in the final version),
- online, interactive benchmarking applications.

Perspectives — future extensions of this work include:

- distributing pre-trained models on Hugging Face,
- adding additional 3D, time-dependent, and multi-physics datasets,
- introducing further evaluation metrics within PLAID.

---

### Decision · Program_Chairs · 2025-09-18

**Decision:**

Reject

**Comment:**

This paper introduces Physics-Learning AI Datamodel (PLAID), a proposed standard/data model for representing physics simulation datasets for the purpose of benchmarking surrogate ML methods for physical systems. The data model is based on the CGNS standard, which is commonly used in computational fluid dynamics (CFD). The authors introduce this standard, along with an open-source Python library, six datasets encoded in this standard (all relevant to CFD), and a benchmark for evaluating common transformer and neural operator models on these datasets.

Reviewers were generally positive about this paper, with no strong concerns noted either before or during the rebuttal period.

I'm concerned about the scale and diversity of the current collection of datasets; it consists of about 1,000 2D simulations which is hardly enough to span the range of interesting CFD phenomena. But I consider the main contribution of the paper to be the PLAID standard (which can be of high impact if there is community adoption), and therefore recommend an accept. I hope that the authors can build upon the initial momentum and ambitiously attempt to expand the scale to that near the current state of the art in physics simulations, such as the Well.

===== FINAL UPDATE FROM DB Track PCs ====

The final decision for this paper has been taken by the program chairs after consultation with the SACs. All Senior Area Chairs have ranked papers according to the feedback from the AC during the review process. We decided to leave the original meta-review to reflect the opinion of the AC in light of the initial discussions with reviewers and SAC.